# TITLE PAGE

## Improved provincial emission inventory and speciation profiles of anthropogenic non-methane volatile organic compounds: a case study for Jiangsu, China

Yu Zhao[1, 2*], Pan Mao[1], Yaduan Zhou[1], Yang Yang[1], Jie Zhang[2, 3], Shekou Wang[3], Yanping Dong[4], Fangjian Xie[5], Yiyong Yu[4], Wenqing Li[5]

1. State Key Laboratory of Pollution Control & Resource Reuse and School of the Environment, Nanjing University, 163 Xianlin Ave., Nanjing, Jiangsu 210023, China

2. Jiangsu Collaborative Innovation Center of Atmospheric Environment and Equipment Technology (CICAEET), Nanjing University of Information Science & Technology, Jiangsu 210044, China

3. Jiangsu Provincial Academy of Environmental Science, 176 North Jiangdong Rd., Nanjing, Jiangsu 210036, China

4. Nanjing Environmental Monitoring Central Station, 175 Huju Rd., Nanjing, Jiangsu 210013, China

5. Nanjing Academy of Environmental Protection Science, 175 Huju Rd., Nanjing, Jiangsu 210013, China

*Corresponding author: Yu Zhao

Phone: 86-25-89680650; Email: *yuzhao@nju.edu.cn*

**ABSTRACT**
Non-methane volatile organic compounds (NMVOC) are the key precursors of
ozone ($O_3$) and secondary organic aerosol (SOA) formation. Accurate estimation in
their emissions plays a crucial role in air quality simulation and policy making. We
developed a high-resolution anthropogenic NMVOCs emission inventory for Jiangsu
in eastern China from 2005 to 2014, based on detailed information of individual local
sources and the field measurements of source profiles of the chemical industry. A total
of 56 NMVOCs samples were collected in 9 chemical plants, and then analyzed with
a gas chromatography-mass spectrometry system (GC-MS). Source profiles of stack
emissions from synthetic rubber, acetate fiber, polyether, vinyl acetate, and ethylene
production, and those of fugitive emissions from ethylene, butanol and octanol,
propylene epoxide, polyethylene and glycol production were obtained. Various
manufacturing technologies and raw materials lead to discrepancies in source profiles
between our domestic field tests and foreign results for synthetic rubber and ethylene
production. The provincial NMVOC emissions were calculated to increase from 1774
Gg in 2005 to 2507 Gg in 2014, and relatively large emission densities were found in
cities along the Yangtze River with developed economy and industry. The estimates
were larger than those from most other available inventories, due mainly to the
complete inclusion of emission sources and to the elevated activity levels from
plant-by-plant investigation in this work. Industrial processes and solvent use were the
largest contributing sectors, and their emissions were estimated to increase
respectively from 461 to 958 and from 38 to 966 Gg. Alkanes, aromatics and
oxygenated VOCs (OVOCs) were the most important species, accounting for
25.9%-29.9%, 20.8%-23.2% and 18.2%-21.0% to annual total emissions respectively.
Quantified with a Monte-Carlo simulation, the uncertainties of annual NMVOCs
emissions vary slightly from years, and the result for 2014 was -41%~+93%,
expressed as 95% confidence intervals (CI). Reduced uncertainty was achieved
compared to previous national and regional inventories, attributed partly to the
detailed classification of emission sources and to the use of information at plant level
in this work. Discrepancies in emission estimation were explored for chemical and
refinery sector with various data sources and methods. Compared with
Multi-resolution Emission Inventory for China (MEIC), the spatial distribution of
emissions in this work were more influenced by the locations of large point sources,
and smaller emissions were found in urban region for developed cities in southern
Jiangsu. Besides, discrepancies were found between this work and MEIC in the
speciation of NMVOC emissions under the atmospheric chemistry mechanisms CB05
and SAPRC99. The difference of species OLE1 resulted mainly from the updated
source profile of building paint use, and the differences of other species from the
varied sector contributions to emissions of the two inventories. CMAQ simulation
was applied to evaluate the two inventories, and better performance (indicated by
daily 1h-max $O_3$ concentrations in Nanjing city) was found for January, April and
October 2012 when the provincial inventory was used.
**1 Introduction**

With strong OH and $HO_2$ radical chemistry reactions, non-methane volatile

organic compounds (NMVOCs) are reported to play crucial roles in formation of
secondary organic aerosols (SOA) and serious photochemical pollution in China,
particularly in developed cities and regions. For example, Huang et al. (2014)
revealed that the contribution of SOA from NMVOC conversion reached 44%-71% to
ambient organic aerosols during a heavy haze period in winter, based on detailed
chemistry composition and source analysis of airborne particles in four important
cities (Beijing, Shanghai, Guangzhou and Xi'an) across the country. Due to intensive
emissions of species with strong atmospheric oxidation capability (indicated as
maximum incremental reactivity, MIR), ozone ($O_3$) formation was recognized as
VOC-limited in developed areas including Jing-Jin-Ji (JJJ), Yangtze River Delta
(YRD) and Pearl River Delta (PRD) regions (Geng et al., 2008; Shao et al., 2009;
Zhang et al., 2008; Xing et al., 2011).
Given the impacts of NMVOCs on air quality, increasing attentions have been
paid to their sources and emission characteristics. Although natural sources dominate
the emissions at global scale (Guenther et al., 1995; Muller, 1992; GEIA,
http://eccad.sedoo.fr/eccad_extract_interface), the contribution from anthropogenic
sources is elevated at smaller spatial scales, attributed to intensive human activities. In
mainland China, emissions of natural and anthropogenic origin were estimated close
to each other at 10-30 Tg, and anthropogenic emissions were dominated by solvent
use and industrial processes (Tie et al., 2006; Klimont et al., 2002; Streets et al., 2003).
Table S1 in the supplement briefly summarizes the estimations of China's national
NMVOC emissions of anthropogenic origin from various studies. With different
methods and data sources applied, NMVOC emissions in China were estimated to be
doubled from 1990 to 2010, and the contributions of solvent use, non-combustion
industrial processes and transportation were enhanced for recent years. Incorporating
available information at national scale, Tsinghua University developed the
Multi-resolution Emission Inventory for China (MEIC, http://www.meicmodel.org/)
and calculated the national total emissions at 23.6 Tg for 2010. Among all the studies,
largest estimations were made in Regional Emission inventory in Asia (REAS, Ohara
et al., 2007; Kurokawa et al., 2013), reaching 28.0 Tg for 2008.
At local scale, emissions from anthropogenic sources could be much higher than
natural sources. For example, the anthropogenic NMVOC emissions were estimated
6-18 times those of natural origin in Beijing (Klinger et al., 2002; Wang et al., 2003;
Klimont et al., 2002; Q. Zhang et al., 2009). With information on individual plants
collected, emission inventories for regions with relatively heavy air pollution in China
including JJJ, YRD and PRD have been developed, and differences in sector
contribution were found. Solvent use and transportation were identified as the largest
NMVOC sources in PRD (Zheng et al., 2009), while industrial processes were more
important in YRD (Huang et al., 2011; Fu et al., 2013). Limitation existed in current
regional inventories. First, the information of local sources was still lacking. Although
combustion sources (e.g., power plants) were gradually included in the regional
emission inventory as point sources, the sources that contribute more to NMVOCs
including refinery and chemical industry plants were less investigated at local scale,
resulting possibly in big bias in emission estimation. Second, with varied data sources
and methods, large discrepancies might exist between studies in the amount and
spatial pattern of emissions. Such discrepancies were rarely analyzed, and the
uncertainties in emission estimation at local scale were seldom quantified. In
particular, the performances of chemistry transport modeling with various NMVOC
inventories have not been sufficiently evaluated. Moreover, source profiles and
speciation of NMVOC emissions need further improvement. Increasingly, domestic
field measurements have been conducted on chemical profiles of NMVOCs for
typical types of sources including solvent use (Yuan et al., 2010; Zheng et al., 2013),
transportation (Tsai et al., 2012; Huang et al., 2015), residential stoves (Wang et al.,
2009), and biomass burning (Kudo et al., 2014). The effects of those results on
speciation of NMVOC emissions were not fully assessed, except for limited studies
(Li et al., 2014). In addition, the measurements on given sectors such as chemical
engineering are still lacking, and the data from foreign countries had to be used.
Under the heavy haze pollution in eastern China (Andersson et al., 2015; Sun et
al., 2015; Wang et al., 2015), series of measures have been conducted particularly on
power and industrial boilers to control the emissions of primary particles and the
precursors of secondary particles such as $SO_2$ and $NO_X$ (Zhao et al., 2014). Along
with gradually reduced ambient PM levels in YRD, $O_3$ pollution becomes a bigger
concern for air quality improvement, motivating the better understanding and
controlling of NMVOC emissions. In this work, we select Jiangsu, a typical province
with intensive refinery and chemical industry in eastern China, to develop and
evaluate the high-resolution emission inventory of anthropogenic NMVOCs. The
geographic location and cities of the province are illustrated in Figure S1 in the
supplement. Field measurements on chemical composition of NMVOC emissions
were conducted to obtain the source profiles of typical chemical engineering
processes. With detailed information of local emission sources collected and temporal
changes tracked, the provincial emission inventory of NMVOC with chemistry
profiles were developed for a ten-year period 2005-2014, and the uncertainties of the
emission estimation were quantified. Through a thorough comparison between results
from varied methods and data sources, the discrepancies in emission estimation,
source profiles, and spatial patterns were then evaluated. Finally, chemistry transport
modeling was applied in southern Jiangsu to test the improvement of the provincial
NMVOC inventory.

**2 Data and methods**
**2.1 Sampling and analysis of NMVOC species from chemical plants**

Chemical profiles of NMVOC emissions are still lacking for chemical industry

and oil exploitation and refinery in China, due to a big variety of source categories.
We select nine types of chemical engineering enterprises that are intensively
distributed in Jiangsu to measure the chemistry composition of NMVOC emissions, as
summarized Table S2 in the supplement. To our knowledge, no domestic
measurement on NMVOC speciation has been conducted for those sources yet, and
current work was expected to supplement the domestic source profiles for chemical
and refinery industry. Based on the on-site investigation of main emission processes,
the locations for stack and/or fugitive emission sampling were determined for each
source type (see Table S2 for the details). Note the sampling cannot be conducted for
all the processes in an enterprise due mainly to the limitation of pipeline layout. The
SUMMA canister produced by University of California, Irvine was employed to
collect the air sample. The canister was made out of stainless steel as its inner wall,
and the volume and vacuum pressure range of the canister was 2 L and $1.1 \times 10^{-4}$ kPa,
respectively. For stack emission measurement, as shown in Figure S2a in the
supplement, the canister was connected with a stainless steel probe, and a filter pipe
filled with glass wool and anhydrous sodium sulfate was applied to remove the
particles and water vapor in the air sample. Under flow control, the sampling time was
roughly 10 minutes until the pressure in the canister reached ambient. For fugitive
emission measurement, the canister was placed 50 meters downwind of the
production device, and the sampling time was roughly 8 minutes. Repeated sampling
was conducted for each process to eliminate the bias and a total of 56 samples were
obtained, as shown in Table S2.
NMVOC samples were analyzed by one Gas Chromatograph Mass Spectrometer
(GC-MS) system (GC6890/MS5973i, Agilent Technologies, USA), as illustrated in
Figure S2b in the supplement. Firstly, the sample was pumped into a cryogenic
pre-concentrator with a three-stage trapping system. In the first stage, the VOC
species were adsorbed on glass beads at $-150°C$ and desorbed at $10°C$. Vapour water
was converted to solid water and was thus separated from sample. In the second stage,
the sample was trapped on Tenax at $-30°C$ and desorbed at $180°C$, and the target
species were separated from $CO_2$ and other compositions in the air. In the third stage,
the sample was focused on a transfer line at $-160°C$ and heated rapidly to $70°C$, and
the species were concentrated. The concentrated VOC was then injected into the gas
chromatograph. The GC oven temperature was initially programmed at $-50°C$, and
then increased to $180°C$ at $4°C/min$ and to $220°C$ at $15°C/min$, holding 3minutes. The
VOC compounds were separated on a DB-5MS capillary column (60 m×0.25 mm×1.0
μm) and quantified using a quadrupole mass spectrometer. The mass spectrometer was
operated in SCAN mode and scanned 20-42 amu and 35-270 amu before and after 8.5
minutes, respectively. The ionization method was electron impacting, and the source
temperature was $230°C$. The PAMS (Photochemical Assessment Monitoring System)
standard mixture (Spectra Gases Inc., USA) and TO-15 standard mixture (Spectra
Gases Inc., USA) were used to confirm the retention times of different compounds,
and to identify them based on retention time and mass spectrum. The target species
were quantified by using multipoint internal calibration method. To establish
calibration curves, a certified gas mixture containing all the target compounds was
dynamically diluted with pure nitrogen to five concentration gradients using mass
flow controllers. Bromochloromethane, 1,2-difluorobenzene, and chlorobenzene-d5
were chosen as internal standards in samples. The method detection limit was 0.5 ppb
for all species.

 **2.2 Development of provincial emission inventory**

Different from national and regional inventories that were based mainly on
energy and economic statistics, detailed information with inter-annual changes for
emission sources in Jiangsu was collected and tracked at plant level from the
multiple-year official Environmental Statistics (the databases of emission sources
compiled by local environmental protection agency), Pollution Source Census (PSC,
internal data of emission sources collected by local environmental protection agency),
and on-site surveys on large emitters conducted by local environmental protection
bureaus. The information included geographic location, types and amounts of raw
materials, types and amounts of products, fuel quality and consumption, and
combustion/manufacturing technology. For 2014, as an example, detailed information
of 6023 plants was obtained, and the locations of those point sources were illustrated
in Figure S1. Incorporating the plant-by-plant information and the energy and
industrial statistics at provincial level, a four-level framework was established,
covering all the anthropogenic NMVOC sources for Jiangsu, as summarized in Table
1. The framework included seven main categories: stationary fossil fuel combustion,
industrial process, solvent use, transportation, oil distribution, biomass burning, and
others. Each main category was further subdivided into subcategories according to
discrepancies in fuel consumption, product manufacturing, or technology application.
The emissions of the province for 2005-2014 were estimated with a bottom-up
method using the following equation:
$$E(n) = \sum_i AL(i,n) \times EF(i,n) \tag{1}$$
where $i$ and $n$ represent the source type and year, respectively; $E$ is the annual
emissions; $AL$ is the activity level data; and $EF$ is the emission factor (i.e., emissions
per unit of $AL$). As summarized in Table 2, emission factors were collected from
extensive literatures and determined as follows with descending priorities: (1) the
results from domestic measurements; (2) the emission limits of local laws and
regulations; (3) the values from expert judgment specific for China; and (4) emission
factors from AP-42 database (USEPA, 2002) and the EMEP/EEA guidebook (EEA,
2013) when domestic information was lacking. Details will be discussed by sector in
Section 2.3.
The total NMVOC emissions for given source type were then broken down into
individual species using Eq. (2):
$$E(i,k) = E(i) \times X(i,k) \qquad (2)$$
where $E$ is the emissions; $i$ and $k$ represent the source type and individual NMVOCs
species, respectively; $X$ is chemical profile of NMVOCs (%). To reduce the
uncertainty of source profile from individual measurement, Li et al. (2014) developed
the "composite profiles" for sources where multiple candidate profiles were available,
by revising the OVOCs fraction and averaging the fractions in different profiles for
each species. As a more detailed source classification was applied in this work, some
sources (e.g., biomass open burning, automobile production & repairing, and wood
decoration & wooden furniture paint) were not covered by Li et al. (2014) and thus
the results from SPECIATE were applied instead. We named the combination of Li et
al. (2014) and SPECIATE as a source profile "before updating" hereinafter. While Li
et al. (2014) included the source profiles published before 2011, a series of local
measurements were conducted after then. In this work, therefore, "composite profiles"
were updated following the method by Li et al. (2014). The most recent domestic
results and the measurments conducted in this work were incorporated as summarized
by source type in Section 2.3.
To evaluate the atmospheric oxidation capability from NMVOC emissions,
ozone formation potentials (OFPs) were calculated by multiplying the speciated
NMVOC emissions and corresponding MIR values (Carter, 1994). To meet the
requirement of atmospheric modeling, NMVOC emissions were assigned to chemical
mechanism (CB05 and SAPRC99) species by multiplying the emissions of individual
species and mechanism-specific conversion factors using the following equation:
$$E(i,m) = \frac{E(i,k)}{M(k)} \times C(k,m)$$

(3)

where $E$ is the emissions, $M$ is the mole weight, $C$ is the conversion factor, and $i$, $m$,

and *k* represent the source type, individual species, and the chemical mechanism species.
The uncertainty of estimated provincial emissions was quantified using a
Monte-Carlo framework (Zhao et al., 2011) for each year. Probability distribution
functions (PDF) were determined for all the parameters involved in the emission
calculation, and 10 000 simulations were performed to estimate the uncertainties of
emissions. The parameters that were most significant in determination of the
uncertainties were identified by source type according to the rank of their
contributions to variance. The detailed information of PDF assumption will be
provided by sector in Section 2.3.
**2.3 Data sources of emission inventory development by category**
**Combustion sources (fossil fuel combustion and biomass burning)**
For power sector and heating/industrial boilers, activity data were compiled at
plant level based on the information obtained from Environmental Statistics, PSC and
on-site survey. The annual amount of residential fossil fuel combustion for 2005-2014
and that of biofuel use for residential stoves until 2008 were directly taken from
Chinese official energy statistics (NBS, 2015a). As the data were unavailable for
subsequent years for biofuel, the activity level was calculated as a product of grain
production, waste-to-grain ratio, and the ratio of residual material burned in stoves
according to the government plan of biomass utilization (PGJP, 2009). The biomass
combusted in open fields was calculated as a product of grain production,
waste-to-grain ratio, and the percentage of residual material burned in the field (Wang
and Zhang, 2008; PGJP, 2009), as described in Zhao et al. (2011; 2013). Following
the rules of IPCC (2006), the uncertainties of activity levels were determined
according to the reliability of energy and economy statistics. As shown in Table S3 in
the supplement, normal distributions were assumed with the coefficients of variation
(CV, the standard deviation divided by the mean) determined at 5%, 10%, 20% and
30% for power, indutrial, residential fossil fuel and biomass combustion sources,
respectively.
As summarized in Table 2, emission factors for power plants and industrial
boilers were taken mainly from Bo et al. (2008). Given the similar designs of boilers
between China and developed countries (Wei et al., 2008), AP-42 database were also
applied when domestic results were lacking. For residential combustion of fossil fuel,
emission factors from domestic tests (Zhang et al., 2000; Wang et al., 2009) were used.
The emission factors for biofuel burning in stoves were from the field test results by
Wang et al. (2009), who measured the NMVOC emission characteristics of multiple
stove-fuel combinations in China and provided the emission factors by province. For
biomass open burning, the local test results by Li et al. (2007) were applied. Chemical
profiles were updated by incorporating domestic measurements for residential fossil
fuel and biofuel burning (Tsai et al., 2003; Liu et al., 2008; Wang et al., 2009; Wang et
al., 2014a), and biomass open burning (Zhang et al., 2013; Kudo et al., 2014). As
measurement results were insufficient for data fitting, uncertainty of emission factor
was evaluated depending on expert judgment for combustion sources (Streets et al.,
2003; Wei et al., 2011). PDF of emission factor was given according to reliability of
data sources and/or the robustness of calculation methods (Wei et al., 2011; also the
case for other sources as indicated below). As summarized in Table S4 in the
supplement, lognormal distributions were assumed with CVs set at 150% and 200%
for fossil fuel and biomass burning, respectively.
**Industrial processes**
Similar with power and industrial combustion, the activity levels of industrial
processes were mainly collected at plant level by source category. However, small
discrepancies existed between the compiled data at plant level and the officially
reported production from economic statistics at provincial level (NBS, 2015b). In
2012, for example, the steel production aggregated from individual plants accounted
for 98% of the provincial total production (Zhou et al., 2017). In this work, therefore,
we treated the individual plants as point sources, and the fraction that was not covered
in plant-by-plant databases as area sources. We assumed the PDFs of industrial
production as normal distribution with CVs at 10% and 20% for point and area
sources, respectively.

Attributed to a wide variety of manufacturing procedures and complicated VOC

exhaust processes, there were few local tests available on emission factors before,
thus the values from expert judgment (i.e., data from routine investigations reported
by the factory officials to local environmental protection bureaus) and data from
AP-42 and EMEP/EEA had to be applied, as summarized in Table 2. Source profiles
of chemical production including rubber, polyether, and polyethylene were obtained
from our measurements described in Section 2.1. Chemical profiles from most recent
domestic measurements were taken for other sources including iron & steel (Shi et al.,
2015; He et al., 2005; Jia et al., 2009; Tsai et al., 2008; Li et al., 2014), paint and
printing ink production (Zheng et al., 2013), and refineries (Liu et al., 2008; Wei et al.,
2014). For sources without sufficient local tests (e.g., food and wine production),
results from foreign studies were applied including the SPECIATE database by
USEPA (2014), and Theloke and Friedirch (2007). Given the potential large
uncertainties, lognormal distribution with CV set at 500% (i.e., a long-tailed PDF)
was assumed for emission factors for most industrial processes.
**Solvent use**

Although solvent-use enterprises were contained in the plant-by-plant surveys,

many of them failed to report the actual solvent usage. Underestimation in activity
levels and thereby emissions could be expected if the information at plant level was
relied on. As the solvent usage was not directly reported at city level, we followed
Wei (2009) to estimate the activity levels for the sector. The total solvent consumption
at national level was first obtained according to the solvent production and imports &
exports statistics (CNLIC, 2015; GAC, 2015). The provincial level was then
calculated based on the intensities of activities that consume solvent (e.g., building
construction and vehicle production). Finally, the provincial data were allocated to
point sources according to production of individual plants and area sources according
to distribution of industrial GDP. Normal and lognormal distributions with CVs set at
20% and 80% were applied for the activity levels of point and area sources of solvent
use, respectively, indicating much larger uncertainty for the latter (Wei, 2009).
The VOC contents of solvent were limited by national laws and regulations, as
summarized by Wei et al. (2008). The updating of regulations and their impacts on
inter-annual changes in NMVOC emission factors were considered from 2005 to 2014,
such as GB18583-2003 and GB18583-2008 for indoor painting. Bias needs to be
admitted here and possible underestimation in emissions would be expected for the
sector, as the regulations were not strictly enforced particularly for small solvent use
enterprises and construction sites (area sources). Data from AP-42, EMEP/EEA and
other literatures (Bo et al., 2008; Fu et al., 2013) were applied when local information
was missing, as provided in Table 2. The domestic tests on chemical species (Yuan et
al., 2010; Zheng et al., 2013; Tang et al., 2014; Wang et al., 2014b) were included to
update the source profiles of the sector. For uncertainty of the emission factors,
uniform distribution was tentatively applied, assuming the same probability in a wide
range (Table S4).
**Transportation**
The activity data of off-road transportation for 2005-2010 was taken from Zhao
et al. (2013), and the data for other years were scaled according to the changes in
passenger and freight traffic by rail and shipping, and those in numbers of agricultural
and construction equipments (NBS, 2015c). For on-road transportation, activity data
(total kilometers traveled) by vehicle type and control stage were calculated as the
product of vehicle population and annual average kilometers traveled (VKT). The
fleet composition by control stage was obtained from the survey by local government
(internal data, Zhao et al., 2015), and VKT by vehicle type were determined according
to previous studies (Cai and Xie, 2007; Wang et al., 2008) and the guidebook of
emission inventory development for Chinese cities (He, 2015). While the CV of
transportation activities at national level was estimated at 16% (Zhao et al., 2011),
larger uncertainties were assumed at provincial level (Wei, 2009). We followed Wei
(2009) and assumed normal and lognormal distribution with CVs at 30% and 50% for
on-road and non-road transportation, respectively.
For emission factors of off-road transportation, the data from EMEP/EEA and
expert judgment (Wei et al., 2008; Q. Zhang et al., 2009) were adopted, attributed to
lack of domestic tests or relevant standards/regulations. Following He (2015), the
emission factors for on-road vehicles were calculated and adjusted according to local
conditions using Eq. (4):
$EF = BEF \times \varphi \times \gamma \times \lambda \times \theta$                                              (4)
where $BEF$ is the base emission factor, $\varphi$ is the environmental correction factor, $\gamma$ is
the average traveling speed correction factor, $\lambda$ is the vehicle deterioration correction
factor, and $\theta$ is correction factor for other conditions (e.g., vehicle loading and fuel
quality). Domestic tests on chemical compositions (Liu et al., 2008; Tsai et al., 2012;
Huang et al., 2015; Wang et al., 2013; Ou et al., 2014; Gao et al., 2012) were
incorporated to update the source profile of on-road vehicles. Lognormal distributions
with CVs at 150% and 300% were respectively assumed for emission factors of
on-road and off-road transportation.
**Oil distribution and other sources**

For oil distribution, Wei (2009) developed a simplified model to simulate the oil

storage, transport and sale activities based on the provincial energy balance statistics,
and the model was applied in this work to calculate the activity levels for the sector.
Activity data from other sources were directly taken from official provincial statistics
(NBS, 2015b; c). Emission factors of oil distribution and other sources were obtained
from Wei et al. (2008), Shen (2006) and Xia et al. (2014), and lognormal distributions
were assumed for them as summarized in Table S4. The source profiles were obtained
from Liu et al. (2008) and SPECIATE.
**3 Results**
**3.1 Source profiles of chemical industry from measurement**

NMVOC source profiles of 14 processes (9 for stack emissions and 5 for fugitive

emissions) were obtained from field measurements. With totally 61 species detected
by GC-MS systems, the chemistry components were grouped into 6 types (alkanes,
alkenes, halohydrocarbon, aromatics, OVOCs, and others), as illustrated in Figure 1a.
Alkanes were the main species of synthetic rubber industry (SBR, SIS rubber and
SEBS rubber) and the mass fractions were measured over 70%. For production of
cellulose acetate fiber, alkanes, aromatics, and OVOCs were the main species in
process of acetate flake production, while OVOCs dominated in the spinning process.
Resulting from various raw materials applied, the source profiles of the two types of
polyether production differed a lot: the mass fraction of OVOCs was close to 80% for
PPG, while the main species for POP were others, aromatics and halohydrocarbon.
For ethylene production, aromatics were the main composition in the stack gas of
cracking furnace, while alkanes and alkenes dominated the fugitive emissions. For
other types of fugitive emissions, big differences existed in the source profiles
attributed mainly to the various raw materials and chemical reactions.
The detailed chemistry compositions for stack and fugitive emissions were
summarized in Tables S5 and S6, respectively. For stack emissions of synthetic rubber
production, cyclohexane was the dominating species, with the mass fractions close to
or above 70% for all types of products. Besides, styrene and acetone were also
important species for SBR and SIS/SEBS rubber, respectively. Used as the solvent in
the chemical reactions, acetone and cyclohexane were considerably emitted during
acetate flake production process, and the mass fraction of acetone reached 70% in the
spinning process. As the raw materials for polyether production, acrylontitrile and
ethylene oxide were the main species emitted from POP and PPG production, with the
mass fractions measured at 43% and 62%, respectively. As the main product,
vinyl acetate was unsurprisingly measured to take 80% of NMVOC emissions from
its production process. For fugitive emissions from ethylene production, the mass
fractions of ethylene, propylene, and n-hexane reached 33%, 26% and 20%,
respectively. 1,2-dichloropropane dominated the emissions from propylene oxide
production, with the mass fraction measured at 65%. For polyethylene and glycol
production, ethylene and xylene were identified as the largest species, with the mass
fractions measured at 42% and 35%, respectively.
Since there were very few domestic tests on source profiles of chemical industry,
the results obtained in this work were compared with those available in SPECIATE
for synthetic rubber, ethylene and polyethylene production, as illustrated in Figure
1b-d, respectively. As can be seen in Figure 1b, large discrepancy was found for
source files of SBR between this work and SPECIATE: while cyclohexane was
identified as the main species in this work, SPECIATE included only styrene and 1,3-
butadiene (the raw materials in SBR production). Variation in manufacturing
technologies was the main source of the discrepancy. Emulsion polymerization
technology was considered in SPECIATE, in which a solvent was not used and thus
NMOVC emissions consisted mainly of the volatile raw materials. This work,
however, measured the plants with solution polymerization technology, in which
significant organic solvents would be released during the drying process. As shown in
Figure 1c, both SPECIATE and our measurements on fugitive emissions indicated that
ethylene and isobutene were the important species for ethylene production. In addition,
much larger fractions of propylene and n-hexane were found in this work. Clear
different composition was found for flue gas of cracking furnace, with abundant
species from incomplete combustion. Similar source profiles were found between this
work and SPECIATE for polyethylene production, dominated by ethylene (Figure 1d).
**3.2 Inter-annual trends and sectoral contribution of NMVOC emissions**
As shown in Table 3, the annual emissions of anthropogenic NMVOCs for Jiangsu
were estimated to increase from 1774 to 2507 Gg during 2005-2014, with an average
annual growth rate at 3.9%. Industrial processes and solvent use were identified as the
largest two sectors contributing to the emissions. The emission fractions of the two
types of sources to total anthropogenic emissions were estimated to increase from
26% in 2005 to 38% in 2014, and from 21% to 39%, respectively. In contrast, the
emission contributions from transportation and biomass burning were declining from
18% to 11%, and from 26% to 4%, respectively, attributed mainly to the controlled
motorcycle emissions, replacement of residential biofuel stoves with natural
gas/electricity ones, and the gradual implementation of straw burning prohibition.
Relatively small contributions were found for stationary fuel combustion plants, oil
distribution, and other sources, and their collective fractions to total emissions ranged
7-9% during the study period.
Figure S3a-c in the supplement provided the inter-annual trends in emissions of
subcategories for industrial processes, solvent use and transportation. The emissions
from industrial processes were estimated to be doubled from 2005 to 2014, and the
inter-annual trend in emissions was well correlated with that in industry GDP (Figure
S3a). The comprehensive investigations on point sources indicated that few measures
were implemented to control NMVOC till 2014, and the increased emissions were
thus mainly driven by the growth of industry activities. Largest growth was found for
synthetic and fine chemical industry, with the emissions elevated from 130 in 2005 to
361 Gg in 2014. Due to enhanced coking industry, the emissions of iron & steel
production were estimated to increase 254% from 27 to 96 Gg.
The emissions of solvent use in Jiangsu were calculated to increase 153% from
380 in 2005 to 963 Gg in 2014, and the growth was highly consistent with that of
industry plus construction GDP (Figure S3b). Despite of increased use of water
soluble paint and implementation of emission standards for given processes (e.g., the
VOC content of interior wall paint has been reduced from 250 to 120 g/kg since 2008),
it was still difficult to restrain the emissions under the fast growth of solvent use, as
relevant polices were not widely conducted across the sector. Paint use was the largest
contributor, and its emissions were calculated to increase from 225 in 2005 to 652 Gg
in 2014. The emissions from printing ink increased 355% from 25 to 115 Gg.
Although Jiangsu's total vehicle population increased 76% from 2005 to 2014,
the NMVOC emissions of on-road transportation were estimated to decline 31% from
297 to 204 Gg, with the peak emissions at 302 for 2007 (Figure S3c). The
implementation of staged emission standards (State I-V, equal to Euro I-V) on new
vehicles, and reduced motorcycle population were the main reasons for the declining
emissions. For example, emissions of motorcycles deceased 66% from 185 to 64 Gg,
and its contribution to on-road vehicle emissions declined as well from 62% and 31%.
Unsurprisingly, gasoline vehicles dominated the emissions of on-road transportation,
with the fraction ranged 81%-87% during the study period.
Illustrated in Figure 2 are the spatial distributions of Jiangsu's NMVOC
emissions for various years within a 3×3 km grid system. The emissions of point
sources were directly allocated according to their geographic locations. For other
sources, certain proxies were applied to allocate emissions, including GDP for
industrial area sources and oil distribution, population for solve use area sources, road
net and traffic flow for on-road vehicles, railway and canal net for off-road
transportation, and rural population for biomass burning. High emission intensities
were mainly found in relatively developed cities along Yangtze River in southern
Jiangsu including Nanjing, Suzhou, Wuxi and Yangzhou (see Table S7 for the
emissions by city). In central and northern Jiangsu, large emissions existed in areas
with clustered industrial parks, reflecting the impacts of big plants on spatial pattern
of NMVOC emissions. Comparing the emissions for 2005 and 2014, increased
emissions were commonly found in southern Jiangsu indicating the faster growth of
industry in developed cities (Figure 2d). Moreover, reduced emissions were coincided
with road net distribution, implying the benefits of emission controls on vehicles.
**3.3 Speciation and OFPs of NMVOC emissions**
Table 4 compares the source profiles of this work with those of Li et al. (2014) or
SPECIATE for typical source categories, grouped as alkanes, alkenes, alkynes,
aromatics, OVOCs, and others. Elevated fractions of alkanes are found in this work
for almost all the sources, while the comparisons of other species are less conclusive
between sources. Based on the source profiles, emissions of more than 500 NMVOC
species were calculated and grouped into 12 categories (alkanes, alkenes, alkynes,
aromatics, alcohols, aldehydes, ketones, ethers, acids, esters, halohydrocarbons, and
others). From 2005 to 2014, the mass fractions of alkanes, unsaturated hydrocarbon
(alkenes and alkynes), aromatics, OVOCs (alcohols, aldehydes, ketones, ethers, acids,
and esters), halohydrocarbons, and others were between 26-30%, 13-19%, 21-23%,
18-21%, 3-4%, and 11-12%, respectively.
Shown in Figure 3 are the mass fractions of species by source for 2014. Due to
varied fuel qualities and combustion conditions, large differences in the speciation of
emissions were found for fossil fuel combustion, transportation and biomass burning.
Dominated by coal combustion, the profile of fossil fuel stationary sources was
relatively simple with little OVOCs and halohydrocarbons, and aromatics were the
largest fraction (45%), followed by alkanes (29%). Alkanes, aromatics and alkenes
were the main species from transportation, with the fractions estimated at 30%, 24%,
and 23%, respectively. For biomass burning, elevated alkenes and less alkanes were
found attributed to the highly incomplete combustion. The mass fractions of alkanes,
aromatics and alkenes from industrial processes were estimated 32%, 16% and 12%.
In particular, relatively close emission fractions were found between species for
chemical industry, the largest emission source of industrial processes: 19%, 15%, 11%,
10% and 10% of aromatics, ketones, alkenes, alkanes, and halohydrocarbons,
respectively. For solvent use, aromatics and alkanes were the most important species
with the fractions estimated at 32% and 22%, respectively, and the collective fraction
of OVOC species reached 27%. Alkanes and aldehydes dominated the emissions of
oil distribution and other sources, which came mainly from the oil evaporation and
residential cooking, respectively.

The OFPs from NMVOC emissions in Jiangsu were calculated to increase from

3880 in 2005 to 5200 Gg in 2014, and the ratio of annual OFPs to emissions varied
slightly around 2.1 for the decade. As the chemical profiles of emitted NMVOC vary
between source categories, the OFP to emission ratio for a given source category
could indicate the potential contribution to ozone formation of the category, as a
combined effect of multiple species emitted. The priorities of emission control for
ozone abatement could thus be suggested by the ratio. The ratios for 2014 were
provided by source in Figure 3. With abundant aromatics and alkenes emissions that
were highly active in chemistry, the largest ratio (3.68) was found for fossil fuel
stationary combustion. The ratios of biomass burning and other sources reached 3.0,
attributed to active aromatics and aldehydes emissions, respectively. The lowest ratio
(1.58) was found for oil distribution, as its emissions were dominated by alkanes with
low reactivity. Figure S4 in the supplement provided 25 species with the biggest
contributions to OFPs and their emission sources for 2005 and 2014. In 2005, the 25
species were estimated to account for 44% of total NMVOC emissions and 83% of
OFPs (Figure S4a). Xylene, ethylene, and propylene were identified as the most three
important species in terms of OFP. The aromatics species with (e.g., xylene and
toluene) came largely from solvent use and industrial processes, while alkenes species
(e.g., ethylene and propylene) were mainly from industrial processes, biomass burning,
and transportation. Besides, biomass burning was the dominating sources of methyl
glyoxal, methyl alcohol and glyoxal. For 2014, the 25 species were estimated to
account for 38% of total NMVOC emissions and 81% of OFPs, and the contributions
of solvent use and industrial processes to OFPs were clearly elevated (Figure S4b).
The orders of isopropanol and n-butanol that were mostly from solvent use, for
example, were moved forward. Moreover, the contribution of biomass burning
significantly declined, and the glyoxal was not identified as the one of the 25 most
important species to OFPs any more.
**3.4 Uncertainties of provincial NMVOC emission inventory**
The uncertainties of estimated annual NMVOC emissions for Jiangsu 2005-2014
are illustrated in Figure 4, expressed as the 95% confidence intervals (CIs) around the
central estimates. As inter-annual changes were hardly assumed in determination of
probability distributions for parameters, similar uncertainty ranges were thus
calculated for emissions of various years. As shown in Table 5, the uncertainty of
emissions 2014 was quantified at -41%~+93% (95% CI), and biomass burning and
other sources were estimated as the sources with largest uncertainties, followed by
stationary fossil fuel combustion and oil distribution. For most emission
sectors/categories, emission factors were identified as the largest sources of emission
uncertainty, with an exception of solvent use. Resulting from complicated sources of
stack and fugitive emissions, it is generally difficult to conduct comprehensive field
tests on emission factors for industrial and residential sources. As described in Section
2.3, large uncertainties had to be conservatively assumed for them due to limited
domestic samples and to heavy dependence on foreign databases. More measurements
on actual emission characteristics are thus recommended to expand data samples for
better evaluating the PDFs and effectively reducing emission uncertainty. Regarding
solvent use for which provincial and city statistics were lacking, the activity data had
to be downscaled from national level leading to possibly big bias in emission
estimation.
Provided in Table 5 as well are the uncertainties of national and YRD emissions
from other studies. Note all the studies included for comparison applied Monte-Carlo
simulation except Huang et al. (2011), which calculated the uncertainty of emissions
based on the predetermined CVs of emission factors and activity levels at sector level.
Compared with those results, the emission uncertainties were reduced in this work for
industrial processes, solvent use and transportation. Besides the varied methods and
assumptions of PDFs for relevant parameters, the more detailed classification of
emission sources and adoption of independent emission factors for those sources
should be an important reason. For example, totally 34 vehicle type-control
combinations were taken into account for calculating the on-road vehicle emissions,
and emission factor for each type of combination was assumed independent from
other. In addition, the errors of activity levels for big point sources were significantly
reduced from the detailed investigation and on-site survey at plant level, leading to
smaller uncertainty in emission estimation for industrial and solvent use sources.
**4 Evaluation of provincial emission inventory**
**4.1 Chemical and refinery industry emissions from varied data sources and**
**methods**
As a part of industrial process, chemical and refinery industry was one of the
biggest contributors to anthropogenic NMVOC emissions. We select Nanjing, the
capital city of Jiangsu province, to evaluate the impacts of data sources and methods
on emissions of this category. As described in Sections 2.2 and 2.3, the method used
in this work for provincial inventory improvement incorporated the most available
information from Environmental Statistics, PSC, and on-site surveys (named Method
1 here). Besides, two other methods based respectively on data from Environmental
Statistics (Method 2) and economic statistics without any information on individual
plants (Method 3, which was commonly applied in national and regional inventories)
were also applied to calculate the emissions in the city for 2011, and the results with
different methods were compared against each other. Note the emissions of area
sources (i.e., not included in plant-by-plant investigations) in Methods 1 and 2 were
estimated using the same data source as Method 3. Table S8 in the supplement
provides the emissions calculated based on the three data sources by subcategory of
chemical and refinery industry. The emissions estimated using Method 1 were clearly
larger than those using Method 2 or 3, resulting mainly from the incomplete records
of chemical products by environmental or economic statistics. For example, some
basic chemistry products (e.g., ethylene oxide and ethylene glycol) and synthetic
chemical products (e.g., polyether and polyethylene) were not included in current
economic statistics. In addition, although most chemical and refinery plants were
investigated in the Environmental Statistics, only three types of chemical products
were recorded for each plant, much less than the actual (more than one hundred types
for some plants). The omission of chemical product types thus led to underestimations
in NMVOC emissions. With the product types fully covered, Method 1 could even
underestimate the emissions, as the emission factors could not be measured or
determined for all products due to the completed pipe layout or fugitive release.
Spatial distributions of the emissions estimated using the three data sources for
Nanjing were illustrated in Figure 5. Similar patterns were found for Method 1 and 2
(Figure 5a and 5b), as the emissions were dominated by the big chemical and refinery
plants. As labeled in Figure 5a, the largest ten plants were estimated to account for
80% of NMVOC emissions of the sector for the whole city. Without detailed
information of individual plants, Method 3 had to apply the proxies (e.g., population
density) to allocate the emissions, and would overestimate the fraction of emissions in
urban downtown (Figure 5c). It could thus be inferred that big discrepancies in spatial
distribution of emissions at small scale might be caused when downscaled from larger
scale without sufficient investigation on local sources, particularly for regions where
emissions were dominated by large plants that were gradually moved out of urban
areas.

To further examine the emission estimation on individual plants, an alternative

method was applied to calculate the emissions of all manufacturing processes
separately including leaks of hydrocarbon vapors from process equipment (valves,
flanges, seals, etc.), storage of organic liquid, loading and unloading of organic liquid,
and waste water treatment (mentioned as device operation based method herein). As
indicated in Table S9 in the supplement, detailed information of 15 key chemical and
refinery enterprises in Nanjing (i.e., enterprises directly under the control of city
government) were collected and the emissions of those plants were calculated and
compared with the results using the method as described in Sections 2.2 and 2.3
(mentioned as emission factor based method herein). Although the total emissions of
the 15 plants were very close between the two methods, significant discrepancies
existed for individual plants. For example, much larger NMVOC emissions were
calculated for plants 3 and 6 with device operation based method, as the emission
factors for production of chlorobenzene alkylbenzene, and cyclohexanone were
lacking, leading to underestimation by the emission factor based method. As shown in
Figure S6 in the supplement, the differences in emissions from varied calculating
methods for the 15 plants led to moderate changes in spatial distributions of the
chemical and refinery emissions for the city. In general, the device operation based
method could better capture the activities of specific plant and the actual emission
characteristics; however, the method could hardly be applied in a broader scale, as it
depends strongly on the completeness and quality of data collection.
**4.2 Changes in speciation of NMVOC emissions**

As indicated in Section 2.2, the speciation of NMVOC emissions in Jiangsu were

updated by including our measurements and other most recent domestic tests after
2010, based on the combination profile of Li et al. (2014) and SPECIATE (i.e., the
source profile before updating). The source types we measured accounted for 9-11%
of annual NMOVC emissions from chemical industry and refinery for Jiangsu
2005-2014. In particular the contribution of those sources was enhanced in typical
cities with intensive chemical industry. In Nanjing, as an example, the source
categories we measured accounted for 19% of annual emissions from chemical
industry and refinery, and for 10% of the total anthropogenic emissions in 2010.
Figure 6 illustrated the emissions of 445 species (accounting for 99.5% of total
NMVOC emissions) estimated with the source profile before and after updating (this
work) for Jiangsu 2010. As a whole, the difference between emissions of all species
before and after updating was calculated at 281 Gg, i.e., 13% of the total
anthropogenic NMVOC emissions for the province. Due to relatively limited tests
available, the changes in emissions were small for most species when updated profiles
were applied. Relative big changes (over 10 Gg) were found for ethylacetate and
certain aromatics species (benzene, xylene, ethylbenzene, and methylbenzene).
Applying the source profile of paint use measured by Zheng et al. (2013) led to
enhanced ethylacetate. Reduced benzene and methylbenzene, and elevated ethyl
benzene and xylene resulted mainly from the revisions on source profiles of cooking
(Jia et al., 2009; Shi et al., 2015) and solvent use (Wang et al., 2014b; Zheng et al.,
2013). Although incremental information on speciation was obtained through the
on-site measurements and source profile updating in current work, domestic data were
still lacking for many source types, given the big variety of source categories for
industrial process and solvent use. Therefore more efforts on field measurments from
different sectors are still needed in order to establish a more complete database of
chemical profiles for the country and the region.
To support the air quality modeling, the emissions of NMVOC species under
CB05 and SAPRC99 mechanisms were calculated using Eq. (4) based on the source
profiles before and after updating, and the results for 2010 were shown in Figure 7
(the speciation for MEIC is illustrated in the figure as well for comparison, as
discussed later in Section 4.3). With source profiles updated based on most recent
measurements, relatively big changes were found for ALDX (see captions of Figure 7
for the detailed meanings) in CB05 and OLE1 in SAPRC99. The revisions on source
profiles of solvent use were the main reason for the changes. For example, the
increased ALDX was attributed mainly to the updated profiles of printing ink and
automobile paint use (Zheng et al., 2013; Tang et al., 2014), while increased OLE1
was to that of building coating (Yuan et al., 2010; Wang et al., 2014b) .

**4.3 Comparisons with other inventories**

The total anthropogenic NMVOC emissions in Jiangsu were extracted from other
continental/national/regional inventories and compared with our estimates for various
years in Figure 4. All the results were within the 95% CIs in this work. Except for
REAS that provided 37-77% higher emissions than this work for 2005-2008
(Kurokawa et al., 2013), our estimates were generally 0-18% larger than other studies
in the total emission estimation, attributed mainly to the omission of certain emission
sources in other inventories and to the elevated activity levels from plant-by-plant
investigation in this work. Figure 8 provided the NMVOC emissions by source from
various inventory studies for selected years. As can be seen, the emissions in this
work were 4% and 20% larger than the national inventory for 2005 (Wei et al., 2008)
and regional inventory for 2010 (Fu et al., 2013), respectively. The latter two studies
missed the emissions from the manufacturing processes of certain chemical products.
For example, fermentation alcohol, dye and rubber were not included in Wei et al.
(2008), either glasswork, pesticide or charcoal in Fu et al. (2013). The emissions from
solvent use in this work were larger than those from Bo et al. (2008), attributed to
omission of carpentry coating, pesticide and adhesive using by the latter. The varied
data sources also contributed to the emission discrepancies. For example, Wei et al.
(2008) and Bo et al. (2008) made larger estimates in transportation emissions than us,
as they applied higher values of annual average miles traveled for motorcycles at
national level.
The emissions of CB05 and SAPRC 99 species estimated by us and MEIC were
compared in Figure 7. While total NMVOC emissions in this work were 315 Gg or
18% larger than MEIC for 2010, relative changes varied among species and could be
bigger for certain ones. In CB05 mechanism, our results were 46% and 43% smaller
for TOL and XYL but 38% and 59% larger for ETH and ETHA than MEIC (Figure
7a), while discrepancies of over 30% (relative to MEIC result) existed for most
species in SAPRC 99 (Figure 7b). Such discrepancies could result either from the
various source profiles, or from the various source contributions in total emissions.
For example, with updated source profile for building coating, much larger OLE1
emissions were estimated in this work than MEIC. Besides the total emissions, the
differed speciations under chemical mechanisms could result in complex impacts on
air quality simulation, which would be discussed in next section.
Figure 9 compares the spatial distributions of Jiangsu's NMVOC emissions for
2010 between our results and MEIC. To be consistent in resolution and to ease
visualization, the high-resolution inventory obtained in this work (Figure 2) was
upscaled to $0.25°×0.25°$, the same as MEIC. Similar spatial patterns were found for
the two inventories: high emission densities existed in southern Jiangsu with
relatively developed economy and industry. As indicated in Figure 9a, the areas with
big plants and large emissions were consistent with each other, indicating that the
provincial NMVOC emissions were largely influenced by the locations of large point
sources. Figure 9c shows that larger emissions estimated in this work than MEIC were
commonly found in areas with big plants, reflecting the impacts of detailed and
complete investigation on product types and activity levels at plant level on the
emission estimation. Although our result was 18% larger than MEIC in total
anthropogenic NMVOC emissions, lower emissions were found in this work at
downtown Nanjing and the Suzhou-Wuxi-Changzhou city clusters with large
populations. The result implies that downscaling of emissions depending on certain
proxies (e.g., population and economy density) might overestimate the emissions in
urban areas, and detailed information on individual sources should be included if
possible.
**4.4 Evaluation of multiple-scale inventories through air quality modeling**
The Models-3/Community Multi-scale Air Quality (CMAQ) version 4.7.1 was
applied to test the performances of chemistry transport simulation with various
NMVOC emission inventories for Jiangsu area. As shown in Figure S6 in the
supplement, three nested domains (D1, D2 and D3) were set, and the most inner D3
covered the mega city Shanghai and six most developed cities in southern Jiangsu
including Nanjing, Changzhou, Zhenjiang, Wuxi, Suzhou and Nantong, with a
horizontal spatial resolution at 3 km. Chemistry transport simulations were conducted
separately with two inventories, i.e., MEIC and the provincial one developed in this
work, for January, April, July and October 2012. Other model settings on
meteorological simulation, chemistry mechanisms and emissions of natural origin
were the same for the two simulations, as described in Zhou et al. (2017). The first
five days for each month were chosen as spin-up period to provide initial conditions
for later simulations.

Figure 10 provides the observed and predicted daily 1h-max $O_3$ concentrations

for the four months, and Table 6 compares the model performances with MEIC and
our provincial inventory, indicated as normalized mean bias (NMB) and error (NME)
values. As suggested by the minus NMBs for most cases, model usually generated
lower 1h-max $O_3$ concentrations than observation with either MEIC or provincial
inventory applied, with an exception for April simulation with MEIC applied. The
result thus implied the updated anthropogenic NMVOC emission inventory at
provincial scale was still likely an underestimation of the actual emissions, as YRD
was commonly recognized as VOC-limited region for $O_3$ formation (Xing et al., 2011).
Compared to MEIC, better model performances (except for July) were generally
achieved when the provincial inventory was applied, indicating the improved
reliability of the detailed bottom-up NMVOC inventory on high-resolution chemistry
transport simulation. In particular, larger emissions were estimated for certain species
with relatively high ozone formation potential (e.g., ethene and ethanol) in the
provincial inventory. It should be noted that the improved ozone simulation was a
combined effect of the updated inventory with revisions on emission estimation,
spatial distribution and source profiles for all the relevant species, and that the impacts
of emission changes for individual species could not be completely validated. More
chemistry transport modeling is further encouraged with intensive sensitivity analysis.

The discrepancies between simulation and observation were still large compared

to regional studies in North America (Y. Zhang et al., 2009). More efforts on
improving or validating emission inventory at provincial scale are thus in great
needed. Besides careful compilation of emission source information in the bottom-up
method, observation constraint from ground measurements could be used to evaluate
the emission level, source contribution, and speciation of VOC emissions (M. Wang et
al., 2014). Emission uncertainty of $NO_X$ could also partly explain the discrepancies, as
the $NO_X$ control measures taken recently could hardly be fully tracked in the emission
inventory development. Besides the limitation of emission input, more analysis on the
impacts of chemical mechanisms and dynamics in the chemistry transport modeling
are also suggested for $O_3$ prediction in the region.
**5 Conclusion**
Using a bottom-up approach, we developed a high-resolution emission inventory
of anthropogenic NMVOC for Jiangsu province, eastern China, with substantial
detailed information on local sources and source profiles from domestic tests
incorporated. Attributed largely to the elevated contribution from industrial processes
and solvent use, the annual provincial emissions were estimated to increase 41% from
2005 to 2014. Influenced largely by location of big point sources, high emission
densities were found in cities along the Yangtze River. Our estimations were larger
than results from most other available inventories except REAS, due mainly to the
omissions of certain industrial and solvent use sources by other studies and to the
elevated activity levels from plant-by-plant investigation in this work. Benefiting from
more detailed classification and investigation of emission sources, reduced
uncertainties in annual emissions were quantified in this work compared to previous
studies. Varied data sources and methods were of significant impacts on emission
estimation at city/plant level. In particular, downscaling directly from national
inventories would overestimate the fraction of emissions in urban downtown. With the
most recent source profiles from local measurements included, considerable changes
in emissions were found for ethylacetate and certain aromatics species, and the
speciation under CB05 and SAPRC99 differed a lot from the national inventory
MEIC. Compared to MEIC, better model performance (indicated by daily 1h-max $O_3$
concentrations) were achieved when the improved provincial inventory was used in
CMAQ simulation, although the discrepancies between simulation and observation
need further investigation. As emission controls on NMVOCs started to be
implemented on industrial sources in China (e.g., the application of leak detection and
repair technique in chemical and refinery plants), more field tests were recommended
to better track the temporal changes in emission factors and to reduce the uncertainty
of emission estimation in the future.
**ACKNOWLEDGEMENT**
This work was sponsored by the Natural Science Foundation of China (91644220
and 41575142), Natural Science Foundation of Jiangsu (BK20140020), Ministry of
Science and Technology of China (2016YFC0201507), Jiangsu Science and
Technology Support Program (SBE2014070918), and Special Research Program of
Environmental Protection for Commonweal (201509004). We would like to
acknowledge Qiang Zhang from Tsinghua University for providing the emission data
(MEIC). Thanks also go to two anonymous reviewers for their very valuable
comments to improve this work.

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

**FIGURE CAPTIONS**

Figure 1. NMVOC source profiles by grouped species measured in chemical enterprises (a) and the comparisons with SPECIATE results for synthetic rubber (b), ethylene (c), and polyethylene production (d).

Figure 2. Spatial distribution of Jiangsu's anthropogenic NMVOC emissions (3×3km) for 2005 (a), 2010 (b), and 2014(c); and the differences between 2005 and 2014 (2014 minus 2005, d).

Figure 3. Fractions of NMVOC species and the ratios of OFPs to emission by sector for Jiangsu 2014.

Figure 4. The inter-annual trends in anthropogenic NMVOC emissions in Jiangsu from 2005 to 2014. The grey dotted lines indicate the 95% CIs of emissions estimated in this study. The results from other inventories are illustrated as well for comparison.

Figure 5. Spatial distributions of NMVOC emissions from chemical and refinery industry for Nanjing 2011 (3×3km), estimated using the detailed enterprise information (a), Environmental Statistics (b), and economic statistics without any information on individual sources (c). The black dots indicate the largest ten emitters in the city

Figure 6. Emissions of NMVOC species for Jiangsu 2010 before and after source profile updating.

Figure 7. Emissions of CB05 (a) and SAPRC99 (b) species for Jiangsu 2010. The left and middle columns indicate the results before and after updating of source profiles, and the right columns indicate the results of MEIC. The CB05 species PAR represents paraffin carbon bond, UNR for unreactive parts of molecules, OLE for terminal olefin carbon bond, TOL for toluene and other monoalkylaromatics, XYL for xylene and other polyalkylaromatics, FORM for formaldehyde, ALD2 for acetaldehyde, ETH for ethene, MEOH for methanol, ETOH for ethanol, ETHA for ethane, IOLE for internal olefin carbon bond, ALDX for C3+ aldehydes, NVOL for nonvolatile mass. The

SAPRC 99 species ALK1, ALK2, ALK3, ALK4 and ALK5 represent alkanes and other non-aromatic compounds that react only with OH, and have kOH between 2 and 500, 500 and 2500, 2500 and 5000, 5000 and 10000, greater than 10000 ppm$^{-1}$ min$^{-1}$, respectively. ETHENE respect ethane, and OLE1 for alkenes with kOH smaller than 70000 ppm$^{-1}$ min$^{-1}$, OLE2 for alkenes with kOH greater than 70000 ppm$^{-1}$ min$^{-1}$, ARO1 for aromatics with kOH smaller than 20000 ppm$^{-1}$ min$^{-1}$, ARO2 for aromatics with kOH greater than 20000 ppm$^{-1}$ min$^{-1}$ and NROG for unreactive mass.

Figure 8. Jiangsu's anthropogenic NMVOC emissions by sector for selected yeas estimated from this work and other inventories. Note emissions in this work are regrouped to be consistent in source categories with Zhang et al. (2009) and MEIC for direct comparison.

Figure 9. Spatial distributions of Jiangsu's anthropogenic NMVOC emissions for 2010 (0.25°×0.25°) by this study (a) and MEIC (b), and the differences between the two inventories (this study minus MEIC, d). The black dots indicate the largest fifty emitters in the province.

Figure 10. Observed and simulated daily 1h-max O$_3$ concentrations using MEIC and provincial emission inventory in January (a), April (b), July (c) and October (d) in 2012 at the air quality monitoring sites in Nanjing. Note the different y-axis scales between panels.

**Table 1 Categories of anthropogenic NMVOC emission sources.**

| Sector | Subsector | Product/fuel/solvent used | Product/technology |
|---|---|---|---|
| Fossil fuel stationary combustion | Power plant | Coal/oil/natural gas/waste/biofuel | N.A. |
| | Heating and industrial boiler | Coal/coke/oil/natural gas | N.A. |
| | Residential | Coal/oil/LPG/natural gas | N.A. |
| Biomass burning | Boiler | N.A. | N.A. |
| | Stove burning | Crop straw/wood | N.A. |
| | Open burning | Crop straw | Rice/corn/wheat/other |
| Industrial process | Iron and steel | Coking | Mechanical/indigenous |
| | | Sinter/pellet/crude steel | N.A. |
| | Non-metallic mineral | Glass | Flat glass/glass fiber/glass work |
| | | Cement clinker/lime/brick and tile/ceramic | N.A. |
| | Oil exploitation and refinery | Crude oil exploit/crude oil refinery | N.A. |
| | Chemical industry | Chemical raw materials: Ethylene/benzene/methanol/ acetic acid/synthesis ammonia/phthalic acid/ethylene oxide/vinylacetate/styrene/glycol/octanol/butanol | N.A. |
| | | Synthetic chemical industry: synthetic resin | Polyethylene/polypropylene/Polystyrene/polyvinylchloride |
| | | Synthetic chemical industry: synthetic fiber | viscose/cellulose acetate fiber/nylon fiber |
| | | Synthetic chemical industry: synthetic rubber | N.A. |
| | | Fine chemical industry: pharmaceutical/chemical pesticide/paint/printing ink/adhesive/dye | N.A. |
| | Food and wine | fermentation alcohol/spirit/beer | N.A. |
| | | Bread/cake/biscuit | N.A. |
| | | Vegetable oil extraction | N.A. |

| Sector | Subsector | Product/fuel/solvent used | Product/technology |
|---|---|---|---|
| Industrial process | Plastic and rubber product | Foam plastic/plastic products/tire/other rubber product | N.A. |
| | Textile industry | Silk/cloth/woolen yarn | N.A. |
| | Carbon black | | N.A. |
| | | | |
| Solvent use | Paint use | Interior/exterior building coating | Water-based/solvent-based |
| | | Automobile production/repairing | N.A. |
| | | Wood decoration/wooden furniture paint | N.A. |
| | | Anticorrosive paint | N.A. |
| | | Other paint | N.A. |
| | Printing | New-type/traditional ink | N.A. |
| | Dyeing | | N.A. |
| | Adhesive use | Shoe making/timber production/other | N.A. |
| | Other solvent use | Agriculture pesticide/Dry cleaning/household solvent use/solvent degreasing | N.A. |
| | | | |
| Transportation | On-road | Automobile | Load (light/heavy)-fuel (gasoline/diesel) |
| | | Motorcycle | Gasoline |
| | Off-road | Train/inland ship/construction machine/ tractor/agriculture truck/agriculture machine | N.A. |
| | | | |
| Oil distribution | Crude oil/gasoline/diesel | Storage/transport/load & unload/gas station | N.A. |
| | | | |
| Others | | Garbage disposal | Burning/landfill/compost |
| | | Cooking fume | N.A. |

**Table 2 NMVOC emission factors for combustion sources, industrial processes and solvent use. Unless specifically noted, the units are g/kg-fuel, g/kg-product and g/kg-solvent, respectively.**

| Source | Fuel/product/solvent used | | EFs in this study | EFs in the references |
|---|---|---|---|---|
| **Combustion** | | | | |
| Fossil fuel | Power plant | Coal | 0.15 | 0.15[a]; 0.81[b]; 0.02[c]; 0.03-0.03[d]; 0.02[e] |
| | | Fuel oil | 0.09 | 3.88[b]; 0.04[c]; 0.12[d]; 0.09-0.14[e] |
| | | Natural gas (g/m$^3$) | 0.083 | 0.12[c]; 0.08-0.10[d]; 0.08-0.10[e] |
| | Heating and industrial boiler | Coal | 0.18 | 0.18[a]; 3.95[b]; 0.04[c]; 0.03-0.03[d]; 0.02[e] |
| | | Fuel oil | 0.12 | 3.88[b]; 0.12[c]; 0.12[d]; 0.09-0.14[e] |
| | | Natural gas (g/m$^3$) | 0.094 | 0.1[c]; 0.08-0.10[d]; 0.08-0.10[e] |
| | Residential | Coal | 4.5 | 0.6[a]; 3.08[b]; 6.48, 4.98[g]; 4.5[c]; 6.41[d] |
| | | Fuel oil | 0.35 | 0.35[c] |
| | | LPG | 5.29 | 0.17[a,b];3.28[g]; 5.29[c]; 66 (g/m$^3$)[e] |
| | | Natural gas (g/m$^3$) | 0.15 | 0.13[c]; 0.15[c]; 0.14[d]; 0.18[e] |
| | | Coal gas | 0.00044 | 0.00044[a] |
| Biomass burning | Boiler | Biomass | 1.1 | 0.0015[e]; 1.1[f] |
| | Stove burning | Wood | 3.23 | 1.09-4.94[g]; 3.23[c]; 5.3[e] |
| | | Crop straw | 13.77 | 1.7- 3.0 (corn straw)[g], 8.89 (wheat straw)[g]; 13.77[c] ; 8.55[d]; 5.3[e] |
| | Open burning | Rice straw | 7.48 | 7.48[h] |
| | | Wheat straw | 7.48 | 7.48[h] |
| | | Corn straw | 10.4 | 10.4[h] |
| | | Other straws | 8.94 | Average of straws open burning above |
| **Industrial process** | | | | |
| Iron and steel | Coking/sintering/steel | | 3.96/0.25/0.06 | 3.4[c]; 3.96[f]/0.25[f]/0.06[f] |
| Non-metallic mineral | Flat glass/glass fiber/glass work | | 4.4/3.15/4.4 | 4.4[c,f]; 3.5[e]/3.15[a, i]/4.4[f] |
| | Cement clinker/lime/brick and tile/ceramic | | 0.33/0.177/0.13/ 29.22 | 0.12[c]; 0.018[d]; 0.33[f]/0.177[f]/0.01[i]; 0.2[c]; 0.033[a]; 0.13[f]/29[c]; 29.215[a]; 29.22[c] |
| Oil exploitation/refinery | | | 1.42/1.82 | 1.42[f]/1.82[f]; 1.05[d]; 3.54[e] |
| Chemical industry | Ethylene/benzene/methanol | | 0.097/0.1/5.95 | 0.097[f]; 0.6[d]/0.1[f]; 0.25[j]/5.95[f] |
| | Polyethylene/polypropylene | | 10/8 | 10[f,j]; 0.33[e]; 3.4[d]/8[f,j]; 0.35[e]; 4[d]; |
| | Polystyrene/polyvinyl chloride | | 5.4/3 | 5.4[f]; 0.21-3.34[e]; 0.12[d]/3[f,j]; 0.33-8.5[e]; 0.096[d] |
| | Acetic acid/ammonia/phthalic anhydride | | 1.814/4.72/21 | 1.81[i]/4.72[f,e]/1.1-6.3[e]; 1.3-6[d]; 21[j] |
| | Ethylene oxide/vinyl acetate/styrene | | 3/4.705/0.223 | 3[f]; 0.98[e]; 2[d]; 4[j]/4.705[i]/0.223[f]; 1[d]; 3.1[j] |
| | Viscose fiber/cellulose acetate fiber/nylon fiber | | 14.5/73.4/3.3 | 14.5[c,f]/73.4[f]; 112[e]/2.13-3.93 |
| | Polyester fiber/acrylic fibers polypropylene fiber | | 0.7/40/37.1 | 0.7[f]; 0.6[e]/3.75-40[e]/37.1[c] |

| Source | Fuel/product/solvent used | EFs in this study | EFs in the references |
|---|---|---|---|
| **Industrial process** | | | |
| Chemical industry | Synthetic rubber | 7.17 | 7.17[f,e]; 3.8-8.6[e]; 0.27-9[d]; |
| | Pharmaceutical/chemical pesticide | 430/20 | 430[f,e]; 300[d]/20[k] |
| | Paint/printing ink/adhesive/dye/pigment | 15/50/30/81.4/10 | 15[f,e]; 11[d]/50[f]; 60[e]/30[c]; 20[f]/81.4[f]/10[i] |
| Plastic and rubber | Foamed plastic /plastic products | 120/3.2 | 770[c]; 120[d]/3.2[f,j] |
| | Tire/other rubber products | 0.91(kg/tire)/12.5 | 0.285(kg/tire)[m]; 0.285(kg/tire)[l]; 0.91 (kg/tire)[f]; 10[d]; 0.659[i]/12.5[c] |
| Food and wine | Fermentation alcohol /white spirit | 60/25 | 218.25[f]; 60[d]/25.35[l]; 25[c,f]; 150(g/L)[d]; 33.105(g/L)[i] |
| | Beer/grape wine | 0.25/0.5 | 0.44[l]; 0.4[c]; 0.43(g/L)[a]; 0.25[f]; 0.35 (g/L)[d]/0.81[l]; 0.5[c,f]; 0.8 (g/L)[d]; 0.38(g/L)[e] |
| | Cake and biscuit | 1 | 1[l,c,d,f] |
| | Vegetable oil extraction | 3.7 | 3.45-10.35[l]; 5.5[c]; 3.45[a]; 1.57(g/L)[d]; 4[i] |
| Textile industry | Silk/cloth/woolen | 10/10/10 | 10[b]/10[b]/10[b] |
| Carbon black | N.A. | 64.7 | 90[m]; 73.2[l]; 64.7[c]; 52[f]; 0.7[d]; 53.57[e]; |
| **Solvent use** | | | |
| Printing | New-type/traditional ink | 100/750 | 100[f];150[c]/750[f]; 650[c],500 |
| Dyeing | - | 81.4 | 81.4[e,f] |
| Paint use | Interior wall painting | 250; 120* | 250[c]/120[n] |
| | Exterior wall painting: aqueous/solvent | 120/580 | 120[f]/360[f];585[c];580[n];230[d] |
| | Vehicle manufacture/refinishing | 470/720 | 460[n]; 470[c]; 475[l]/720[c,d]; 850[l] |
| | Furniture: aqueous/solvent based | 250/670 | 250[c]/640[f]; 637[n]; 550-750[c]; |
| | Anticorrosive paint use | 442 | 442[c] |
| | Other paint use | 240 | 240[c] |
| Adhesive use | Shoe making | 670 | 664[n]; 670[c] |
| | Timber production | 90 | 88[n]; 90[c] |
| | Other adhesive use | 89 | 89[c]; 85[l] |
| Other solvent | Agriculture pesticide | 470 | 368-482[c]; 356-576[f] |
| | Dry cleaning | 0.16** | 0.8[e]; |
| | Household solvent use | 0.08 | 0.08[f]; 3.7[d]; 4.2[e]; 0.1[a] |
| | Degreasing | 0.044 | 0.044[a,f] |

[a]Bo et al. (2008); [b]Tang and Chen (2002); [c]Wei et al. (2008), Wei(2009), and Wang et al.(2009); [d]EEA (2013); [e]USEPA (2002); [f]MEP (2014); [g]Zhang et al. (2000); [h]Li et al. (2007); [i]ROC EPA (2009); [j]Fan et al. (2012); [k]Xia et al. (2014); [l]Wang (2006); [m]Klimont et al. (2002); [n]Fu et al. (2013)

*250 and 120 g/kg for 2005-2007 and 2008-2014, respectively; **Adjusted by per capital income.

**Table 3 Anthropogenic NMVOC emissions in Jiangsu by source from 2005 to 2014 (Gg).**

| Source | 2005 | 2006 | 2007 | 2008 | 2009 | 2010 | 2011 | 2012 | 2013 | 2014 |
|---|---|---|---|---|---|---|---|---|---|---|
| Fossil fuel combustion | 43 | 43 | 45 | 47 | 48 | 50 | 54 | 54 | 58 | 59 |
| Industrial process | 461 | 580 | 616 | 626 | 722 | 823 | 747 | 821 | 871 | 958 |
| Transportation | 325 | 321 | 336 | 331 | 331 | 317 | 313 | 302 | 311 | 280 |
| Solvent use | 380 | 441 | 492 | 495 | 606 | 652 | 697 | 783 | 809 | 966 |
| Oil distribution | 35 | 35 | 39 | 42 | 46 | 53 | 56 | 64 | 59 | 53 |
| Biomass burning | 458 | 343 | 355 | 147 | 139 | 131 | 126 | 119 | 116 | 110 |
| Other | 72 | 73 | 74 | 74 | 75 | 76 | 77 | 78 | 78 | 79 |
| Total | 1774 | 1835 | 1956 | 1762 | 1967 | 2102 | 2071 | 2220 | 2302 | 2507 |

**Table 4 Updated NMVOC source profiles in this study and those from SPECIATE and Li et al. (2014).**

| Sector | | Species (weight percentage, %) | | | | | |
|---|---|---|---|---|---|---|---|
| | | Alkanes | Alkenes | Alkynes | Aromatics | OVOCs | Others |
| Biomass open burning | This study | 14.83 | 17.24 | 2.20 | 10.98 | 52.58 | 2.16 |
| | SPECIATE | 14.20 | 20.59 | 2.62 | 1.82 | 57.68 | 3.08 |
| Coking | This study | 25.65 | 25.00 | 1.34 | 25.42 | 1.43 | 21.16 |
| | SPECIATE | 9.17 | 39.45 | 2.14 | 49.24 | 0.00 | 0.00 |
| Iron& steel | This study | 25.64 | 11.23 | 15.56 | 13.60 | 2.36 | 31.62 |
| | SPECIATE | 28.06 | 16.67 | 27.72 | 0.00 | 0.00 | 27.56 |
| Paint production | This study | 3.90 | 0.00 | 0.00 | 48.61 | 42.76 | 4.72 |
| | SPECIATE | 0.00 | 0.00 | 0.00 | 80.65 | 19.35 | 0.00 |
| Ink production | This study | 54.48 | 15.27 | 0.00 | 30.25 | 0.00 | 0.00 |
| | SPECIATE | 5.00 | 0.00 | 0.00 | 6.50 | 63.00 | 25.50 |
| Refinery | This study | 77.02 | 12.10 | 0.00 | 10.89 | 0.00 | 0.00 |
| | Li et al., 2014 | 67.57 | 9.44 | 0.91 | 1.96 | 0.00 | 20.12 |
| Furniture painting | This study | 6.15 | 0.14 | 0.00 | 61.61 | 29.23 | 2.87 |
| | SPECIATE | 0.00 | 0.00 | 0.00 | 30.00 | 51.42 | 18.58 |
| Architecture painting | This study | 28.69 | 9.39 | 0.00 | 61.92 | 0.00 | 0.00 |
| | Li et al., 2014 | 27.48 | 4.71 | 0.00 | 67.81 | 0.00 | 0.00 |
| Vehicle varnish paint | This study | 3.65 | 0.55 | 0.06 | 59.09 | 32.02 | 4.64 |
| | Li et al., 2014 | 2.14 | 0.82 | 0.08 | 96.96 | 0.00 | 0.00 |
| Printing | This study | 29.54 | 2.90 | 0.69 | 16.64 | 44.89 | 5.33 |
| | Li et al., 2014 | 14.78 | 2.79 | 0.66 | 11.03 | 26.96 | 43.78 |

| Sector | | Species (weight percentage, %) | | | | | |
|---|---|---|---|---|---|---|---|
| | | Alkanes | Alkenes | Alkynes | Aromatics | OVOCs | Others |
| Diesel vehicle | This study | 44.30 | 22.41 | 1.79 | 20.10 | 11.39 | 0.00 |
| | Li et al., 2014 | 14.88 | 11.60 | 0.83 | 8.99 | 48.99 | 14.71 |
| Motorcycle | This study | 41.36 | 23.36 | 2.11 | 28.50 | 4.68 | 0.00 |
| | Li et al., 2014 | 45.88 | 32.18 | 0.56 | 21.37 | 0.00 | 0.00 |

**Table 5 Uncertainties of anthropogenic NMVOC emissions (expressed 95% CI around the central estimates) and the most significant two parameters contributing to the uncertainties by sector for Jiangsu 2014. The percentages in the parentheses indicate the contributions of the parameters to the variance of emissions. The uncertainties from other inventories are provided for comparisons.**

| Source Category | Uncertainty | | | | | Parameters contributing most to uncertainty | |
|---|---|---|---|---|---|---|---|
| | This work | Wei (2009) | Bo et al. (2008) | Fu et al. (2013) | Huang et al. (2011) | This work | |
| | Provincial scale | National scale | National scale | Regional scale | Regional scale | Provincial scale | |
| Fuel stationary combustion | -66%, +190% | - | - | - | - | $EF_{Power\ plant,\ coal}$ (68%) | $EF_{Industrial\ boiler,\ coal}$ (6%) |
| Industrial process | -58%, +152% | -88%, +283% | - | -57%, +152% | -60%, +152% | $EF_{Tire}$ (23%) | $EF_{Coking}$ (13%) |
| Solvent use | -68%, +131% | -82%, +223% | - | -60%, +147% | -59%, +150% | $AL_{External\ wall\ paint}$ (20%) | $EF_{Other\ paint}$ (14%) |
| Transportation | -51%, +117% | -86%, +261% | - | - | - | $EF_{Inland\ ship}$ (13%) | $EF_{Construction\ machine}$ (8%) |
| Oil distribution | -66%, +162% | - | - | - | - | $EF_{Crude\ oil\ storage}$ (27%) | $EF_{Gasoline\ sale}$ (23%) |
| Biomass burning | -76%, +499% | - | - | - | - | $EF_{Straw-stove}$ (74%) | $R_{Straw\ burning\ in\ stove}$ [1] (5%) |
| Other | -98%, +490% | - | - | | - | $EF_{Cooking}$ (84%) | $EF_{Garbage\ burning}$ (14%) |
| Total | -41%, +93% | -51%, +133% | -36%, +94% | -52%, +105% | -53%, +113% | | |

[1] The ratio of straw burned in stove as biofuel

**Table 6 Model performance of daily 1h-max O₃ concentrations using MEIC and provincial inventory for January, April, July and October 2012 in Nanjing.**

| | Provincial emission inventory | | MEIC | |
|---|---|---|---|---|
| | NMB[1] | NME[1] | NMB | NME |
| January | -21% | 34% | -58% | 59% |
| April | -26% | 38% | 35% | 55% |
| July | -28% | 33% | -23% | 29% |
| October | -20% | 26% | -50% | 50% |

[1] Normalized mean bias (NMB) and error (NME) were calculated as following equations ($P_i$ and $O_i$ indicate the results from modeling prediction and observation, respectively):

$$NMB = \frac{\sum_{i=1}^{n}(P_i - O_i)}{\sum_{i=1}^{n}O_i} \times 100\% \quad NME = \frac{\sum_{i=1}^{n}|P_i - O_i|}{\sum_{i=1}^{n}O_i} \times 100\%$$

;

**Figure 1**

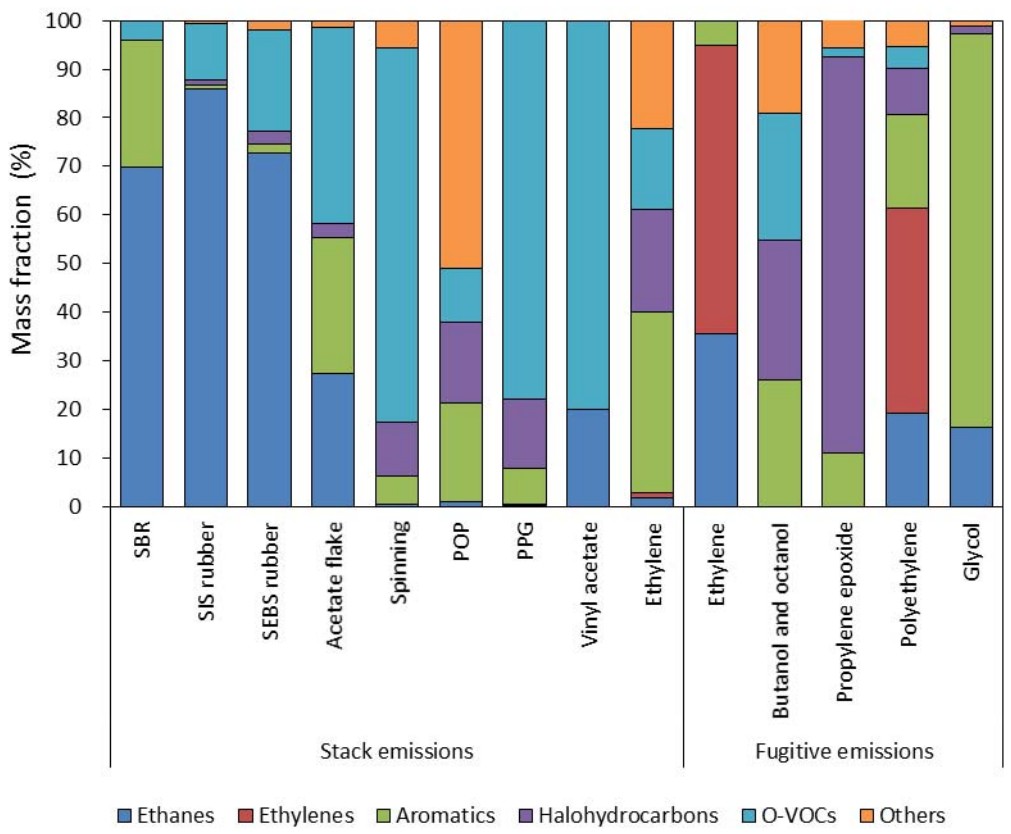

(a)

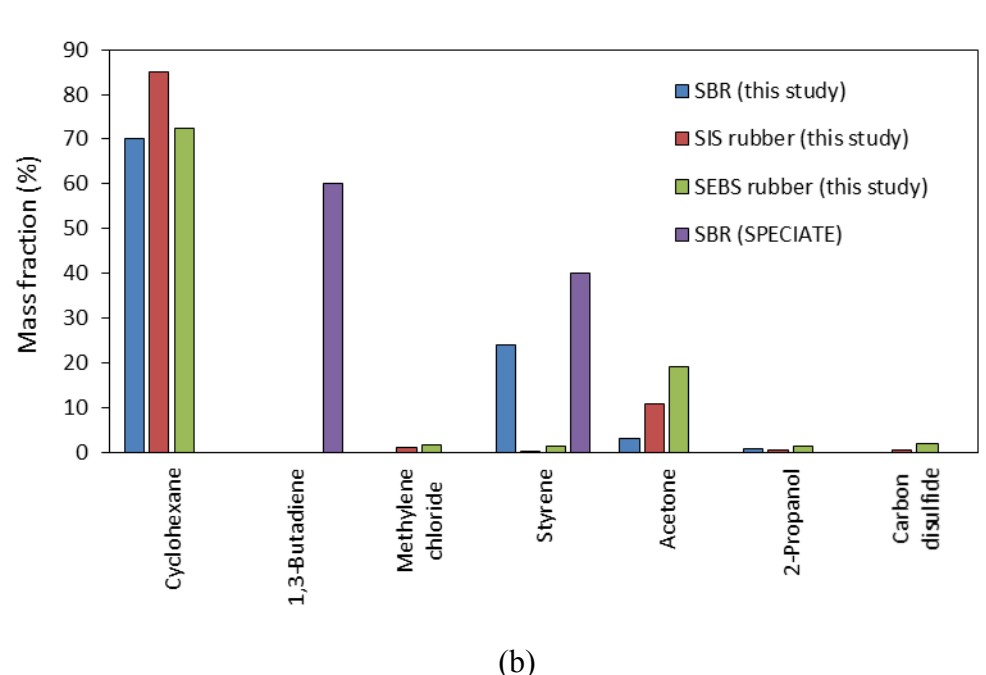

(b)

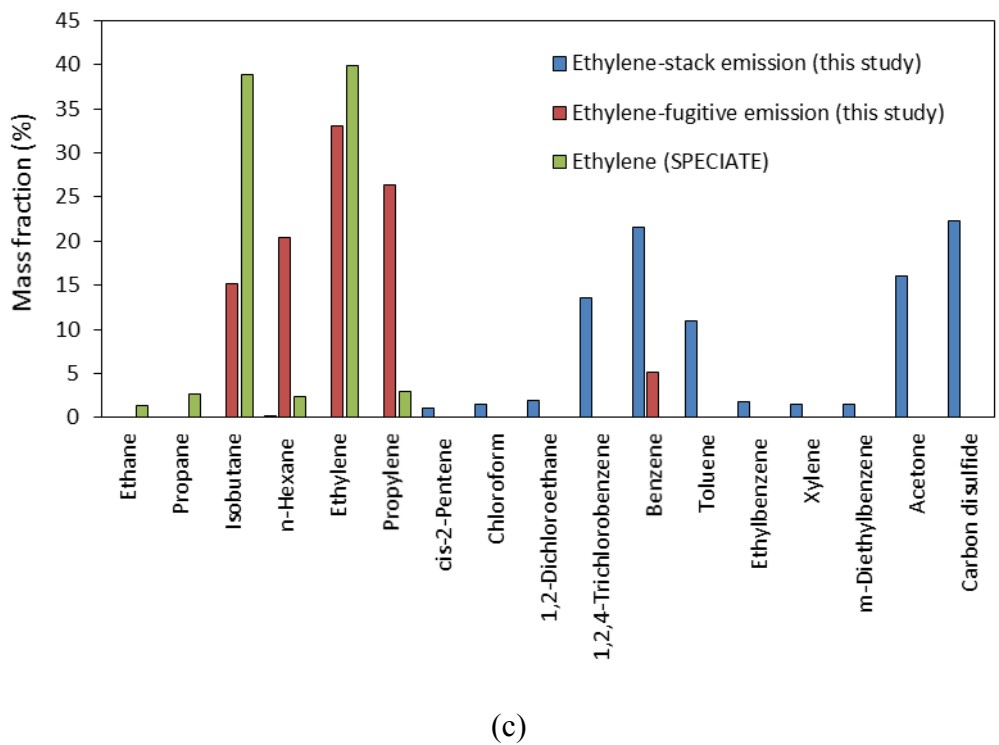

(c)

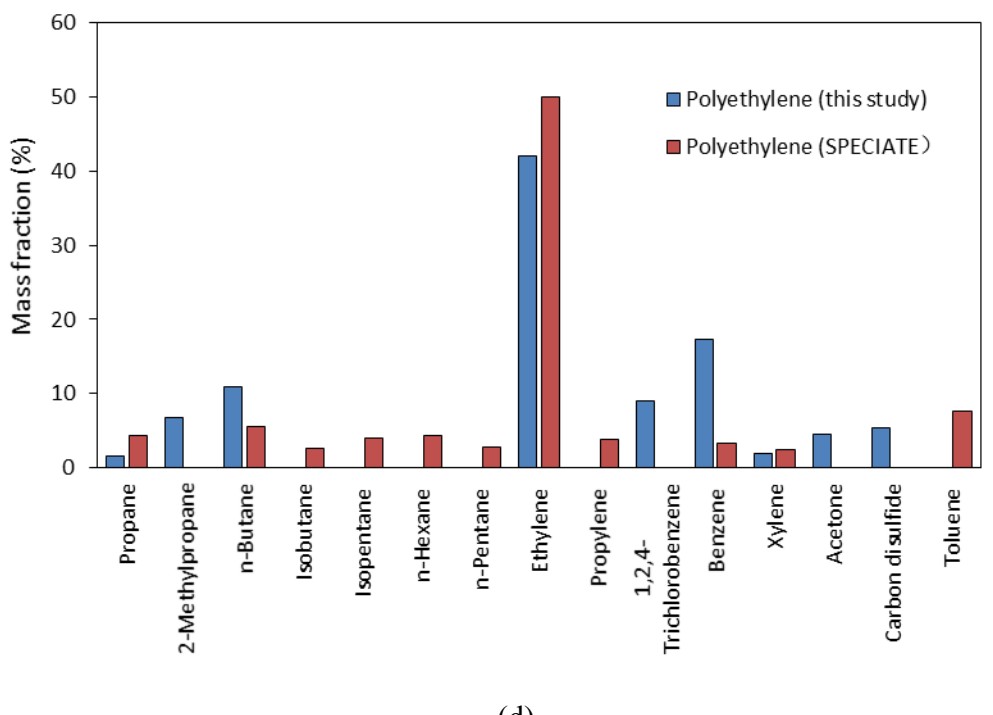

(d)

**Figure 2**

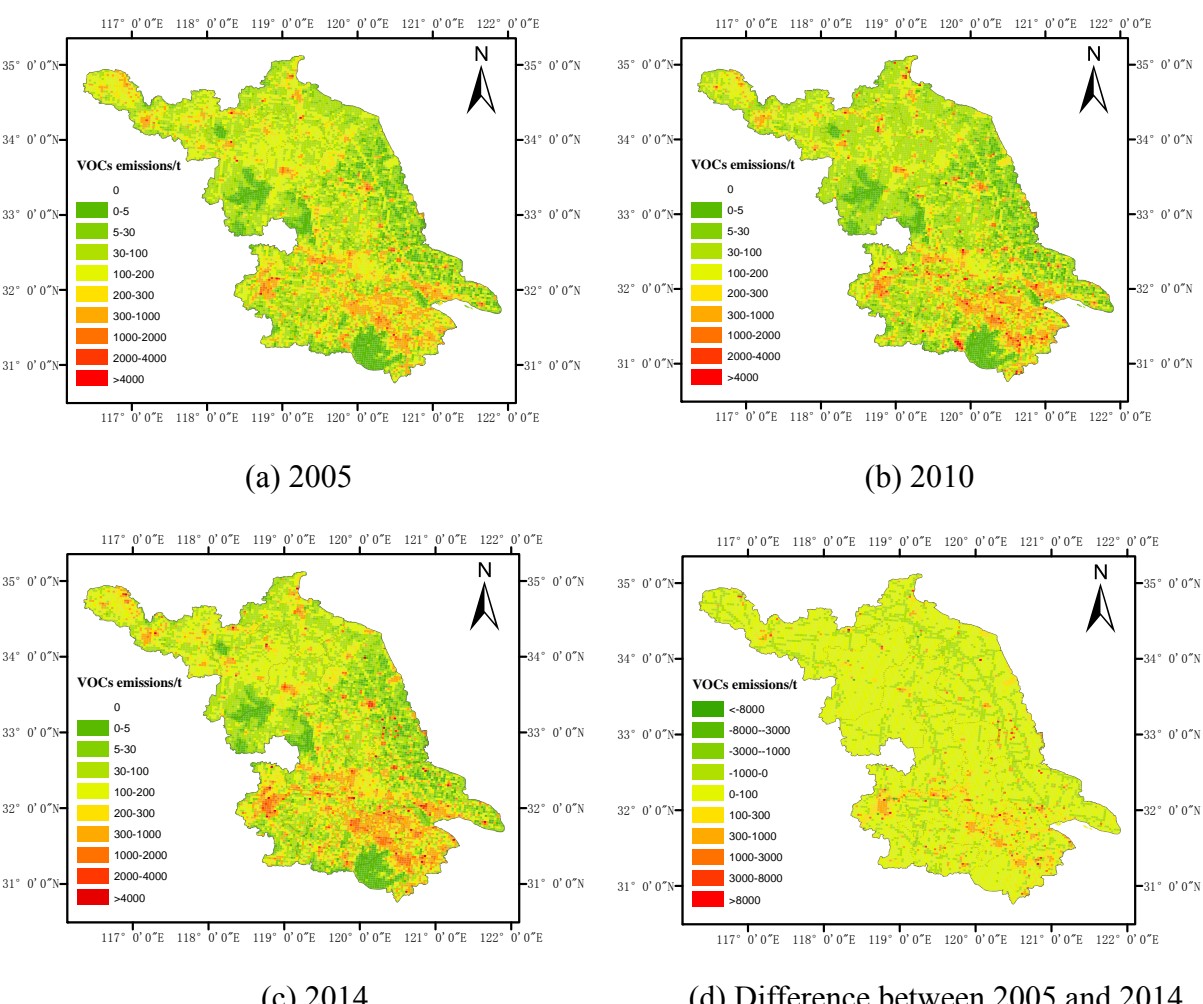

(a) 2005                                    (b) 2010

(c) 2014                    (d) Difference between 2005 and 2014

**Figure 3**

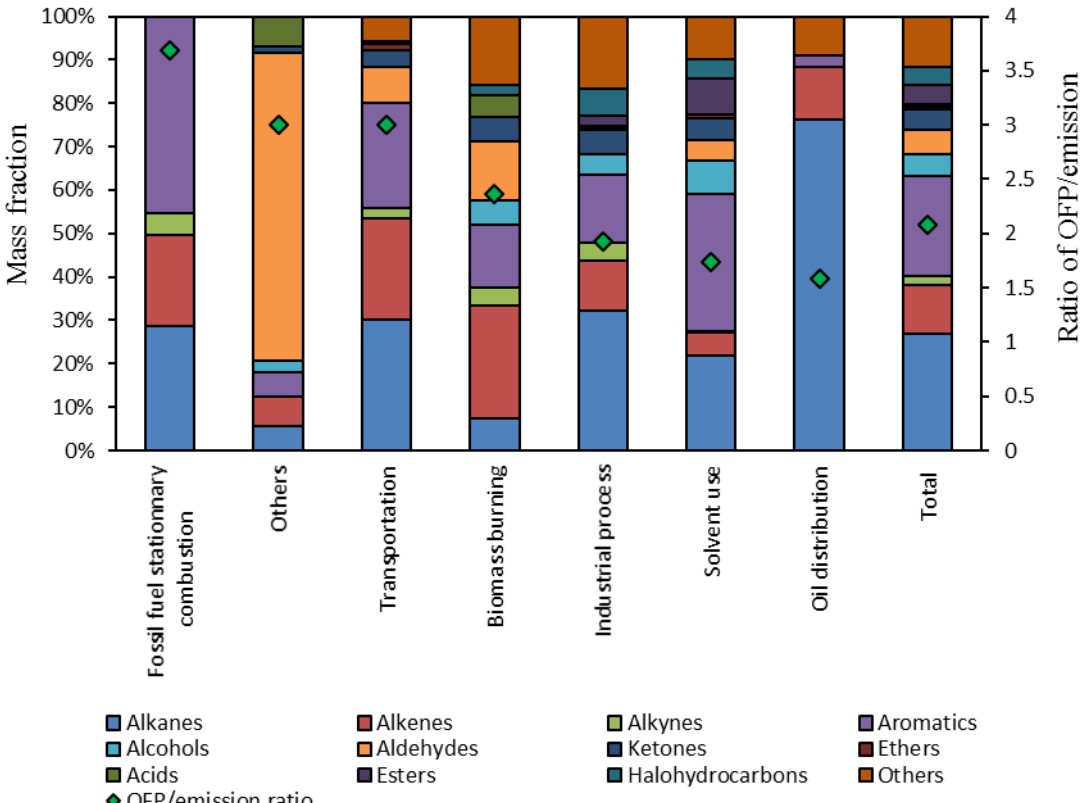

**Figure 4**

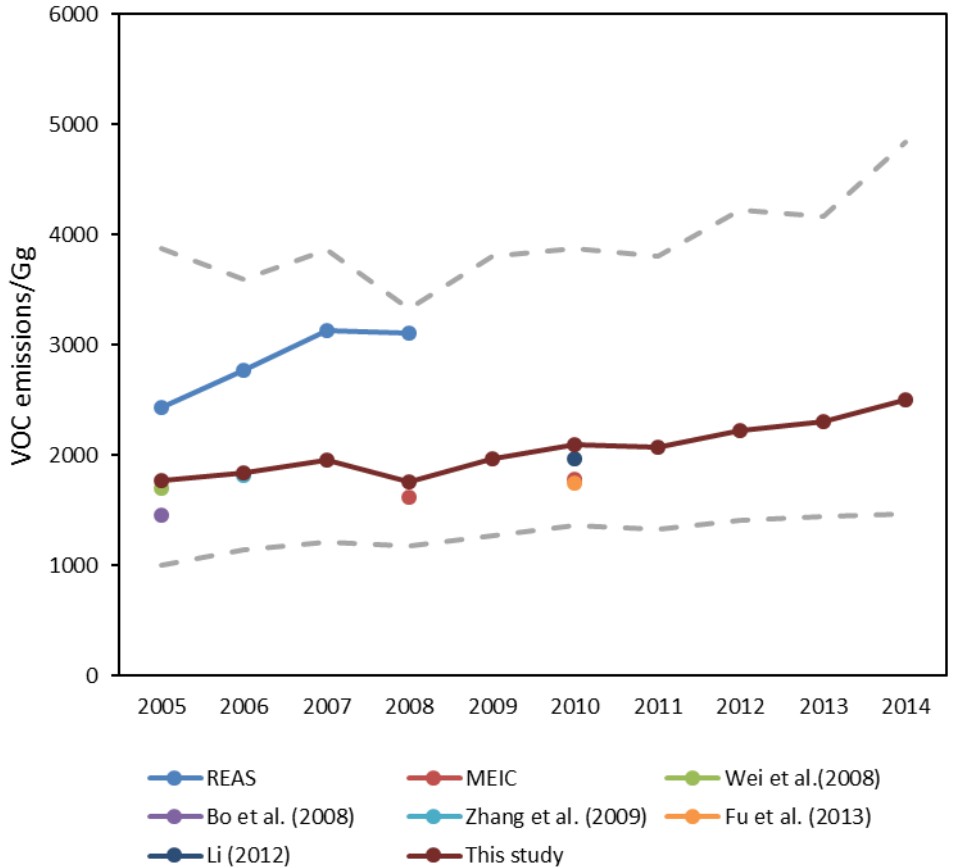

**Figure 5**

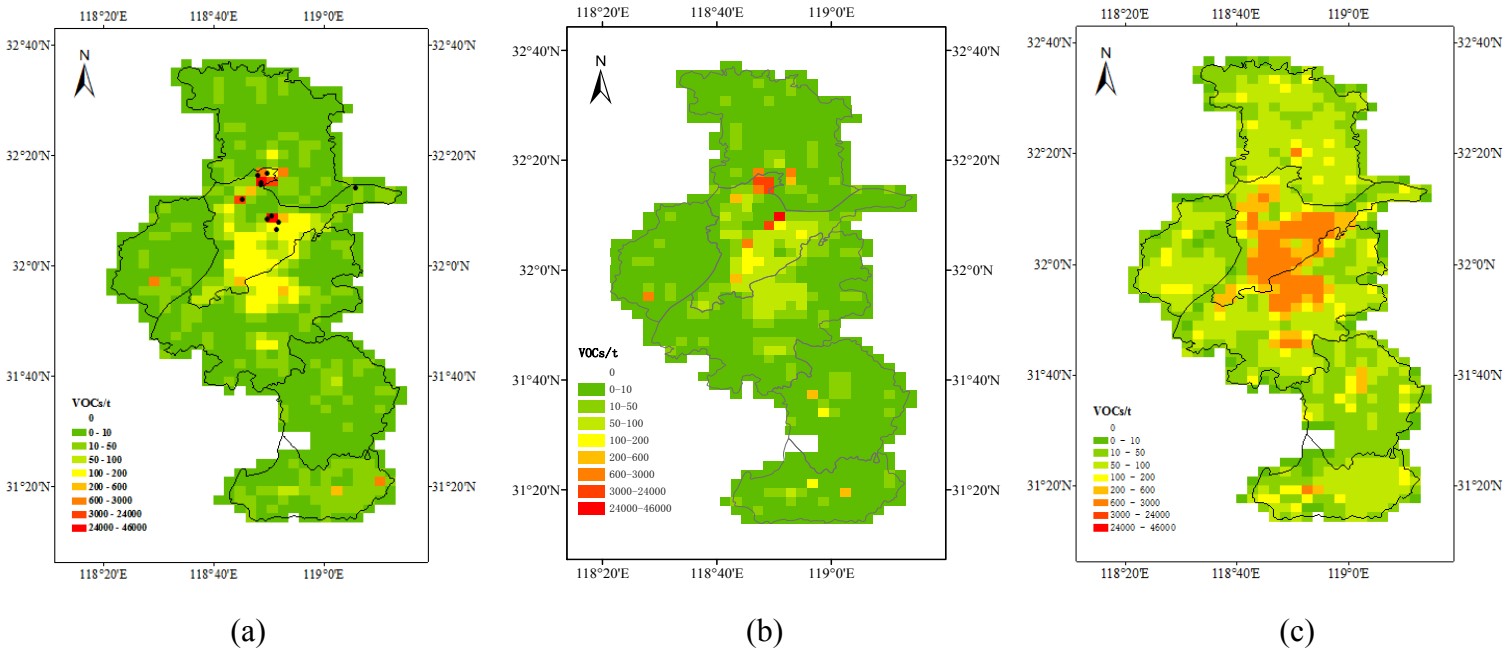

(a)                    (b)                    (c)

**Figure 6**

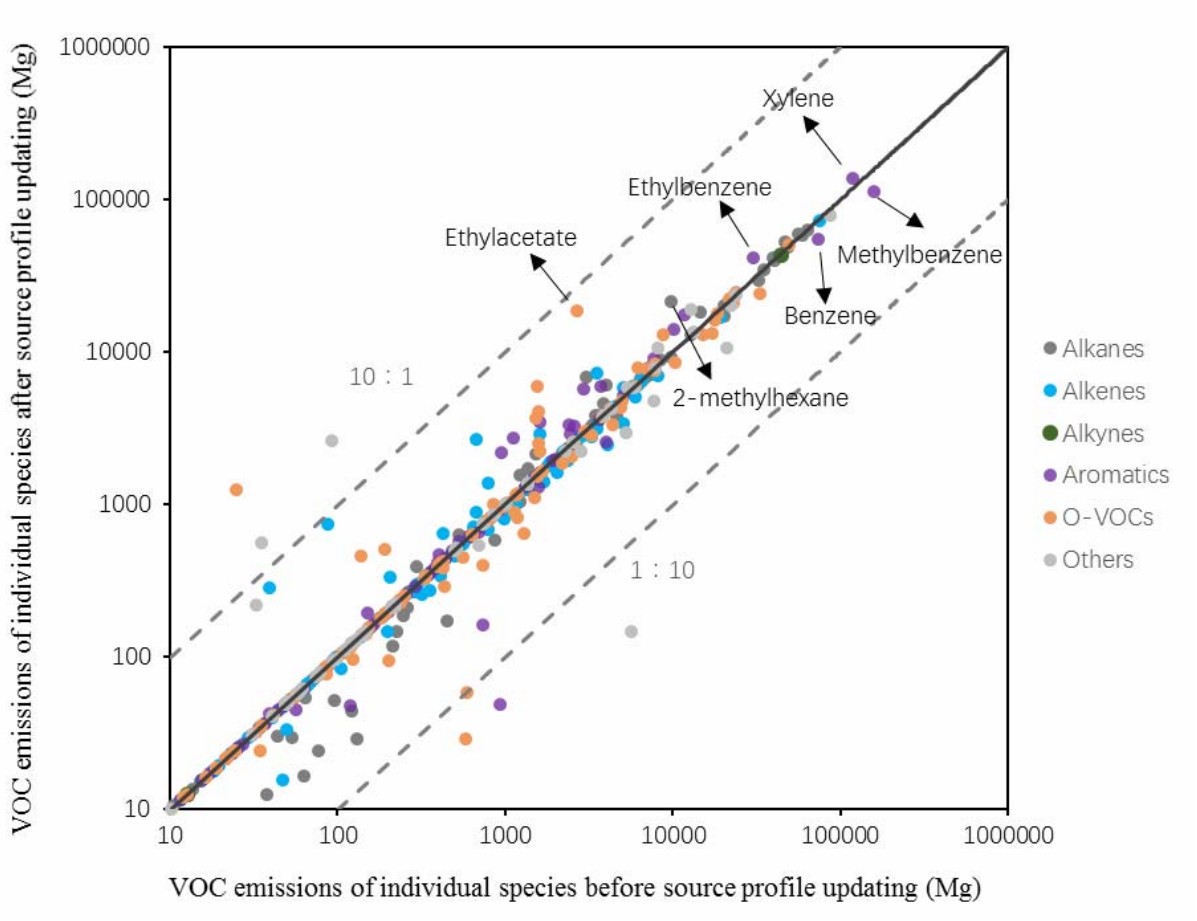

**Figure 7**

(a) CB05

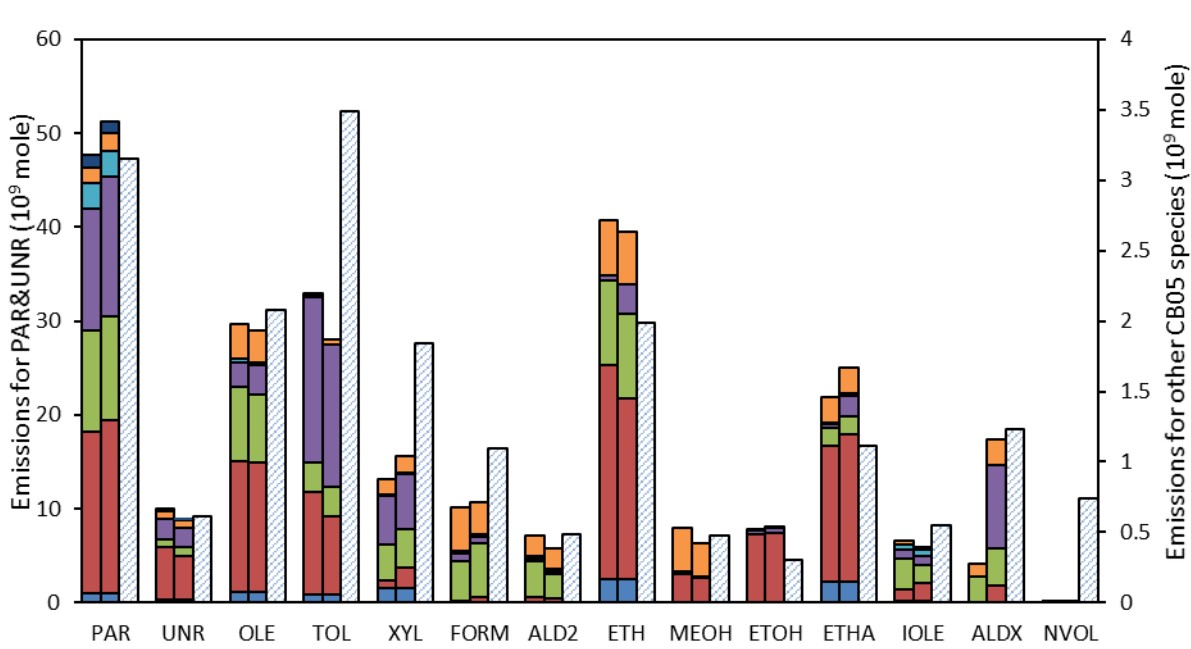

(b) SAPRC99

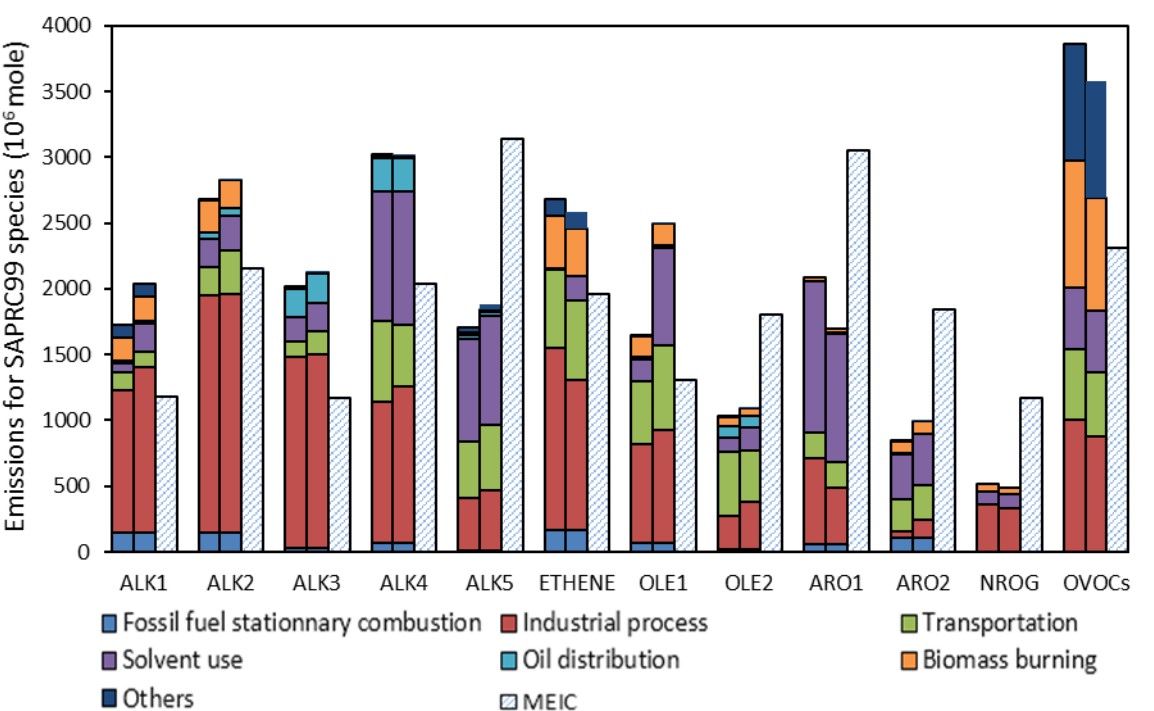

**Figure 8**

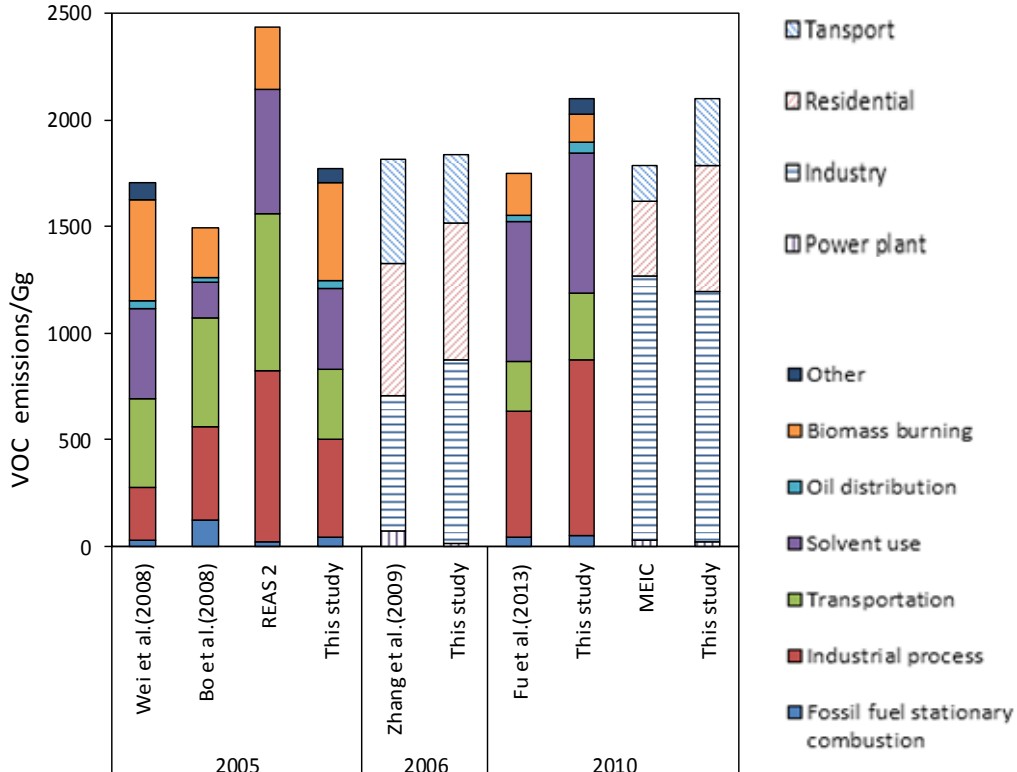

**Figure 9**

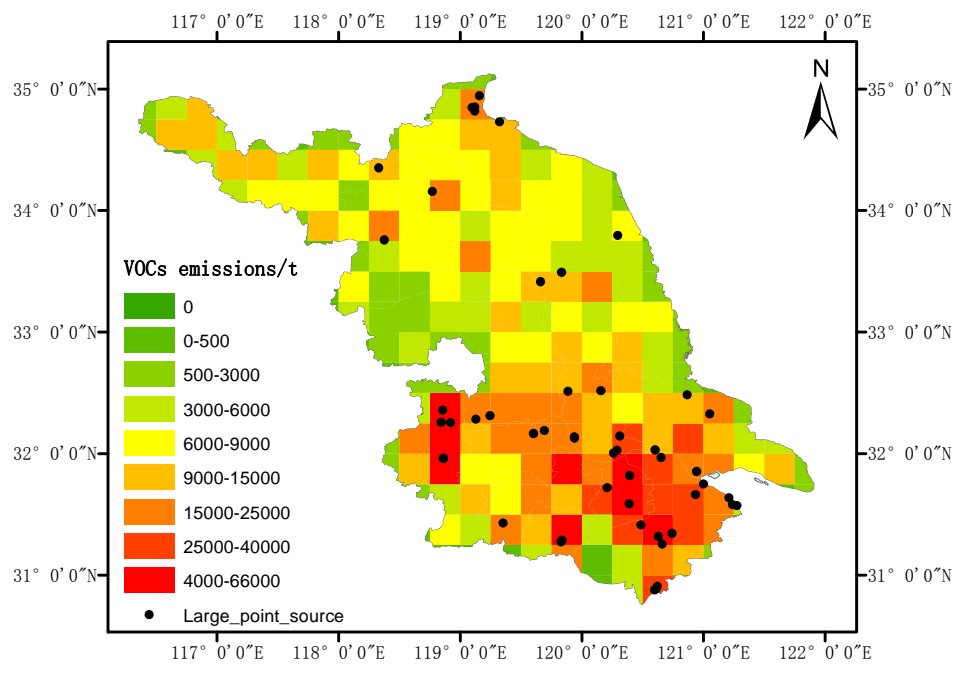

(a)

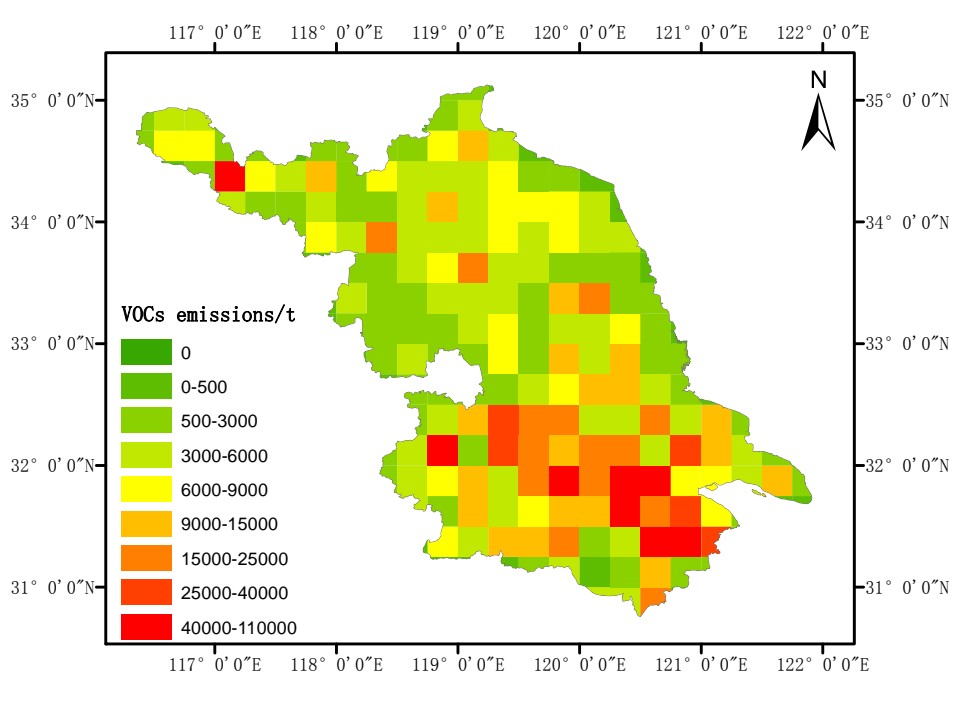

(b)

Figure 9 (continued)

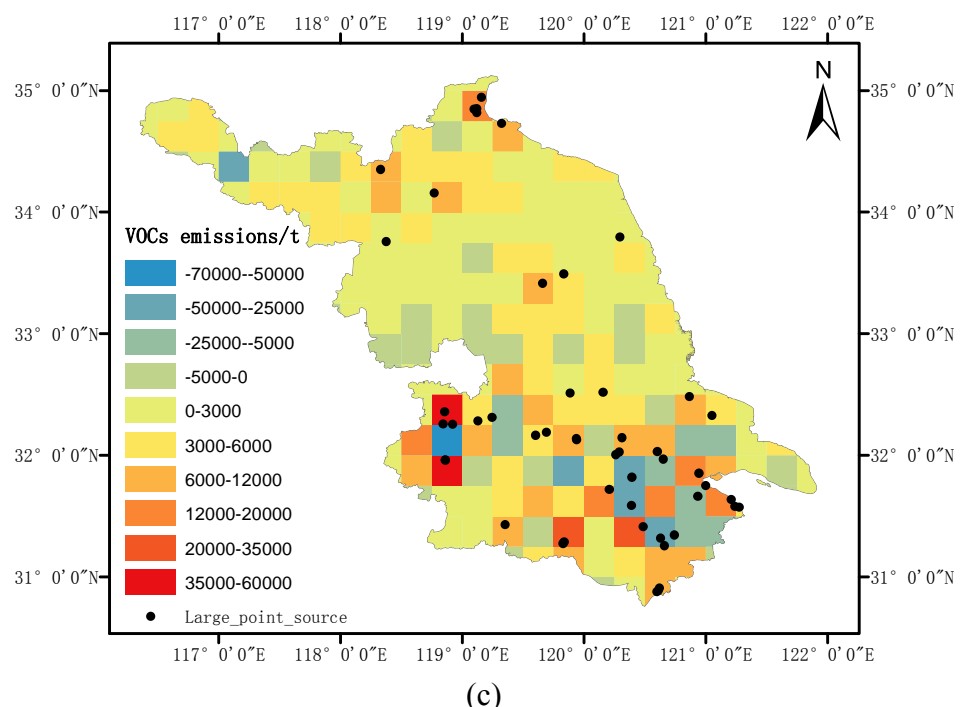

(c)

**Figure 10**

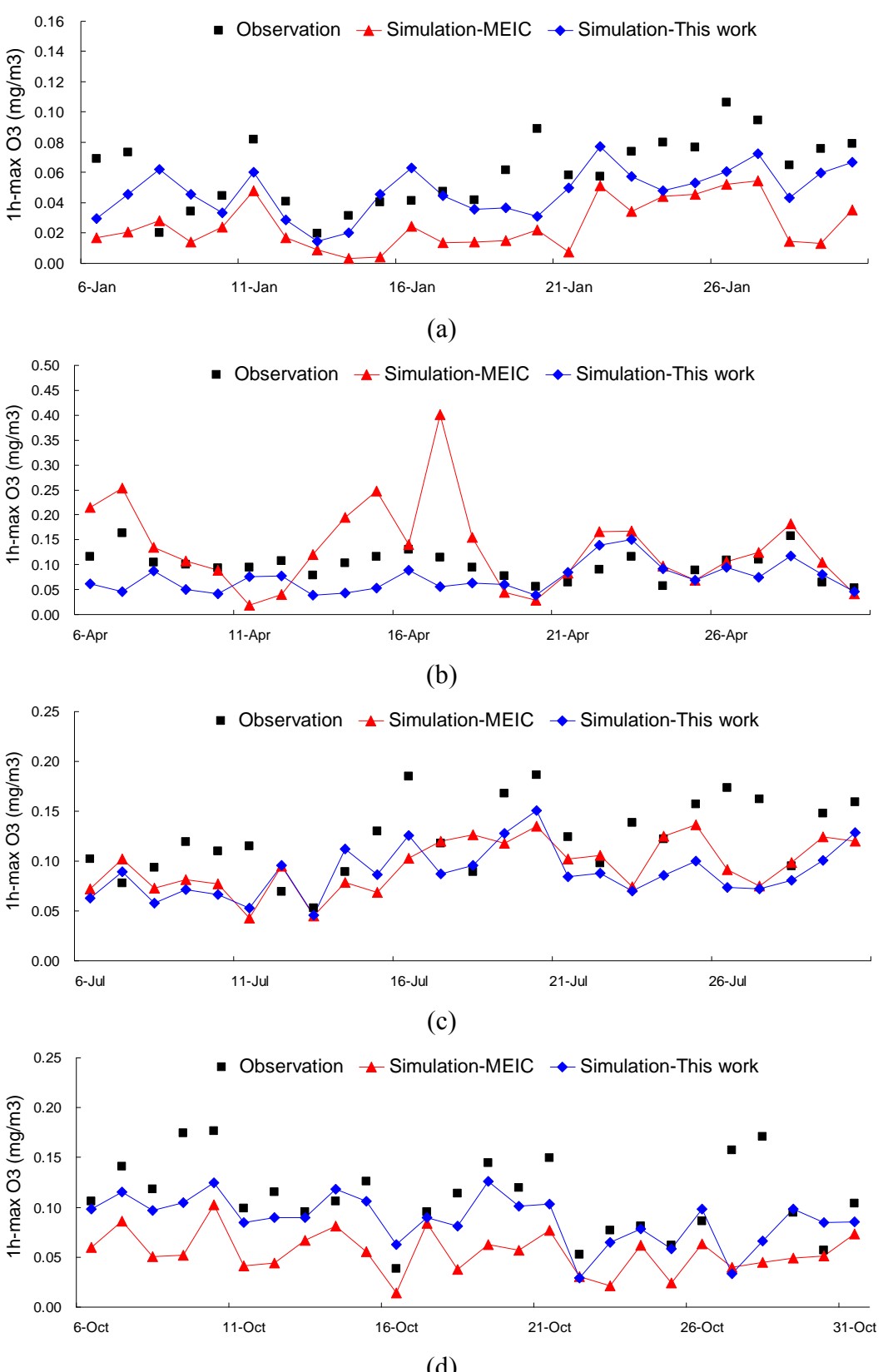

(a)

(b)

(c)

(d)