# Peer review of "Improved provincial emission inventory and speciation"

_Atmospheric Chemistry and Physics, 2016_

## Referee Comment (RC1) · Anonymous Referee #1 · 10 Mar 2017

This paper describes the development and updates of a bottom-up regional anthropogenic emissions inventory for the Jiangsu province in China. Source profiles of industrial facilities were improved compared to previous inventories through measurements at the facility level of NMVOCs from canister samples. Differences to other inventories are discussed and regional CMAQ modeling studies with the various inventories were performed and it was found that using the current inventory improves, albeit still underestimates, ozone predictions. Improving anthropogenic emission inventories is the main pathway to improving air quality and climate modeling and forecasting, especially in areas such as China, which makes this paper a very valuable contribution.

Unfortunately, the paper is difficult to read because of necessary information hidden in the SI, important data sources missing from the description and generally inconsistent description of the different source types and as a result this paper needs major changes, before it is acceptable.

General Comment:

My main issue with this paper is that the description of the inventory development in chapters 2.2 and 2.3 is pretty unclear. First of all, the essential information of the source types is in the SI and not in the main text. A paper needs to be understandable even without reading the SI in detail and that is certainly not the case here. Tables S1 and S3 should be combined into one and moved to the main text. I also think that Tables S4, S5 and S6 should be combined and moved to the main text. Then the source types and the data sources for each type need to be discussed in more detail.

I would like to see a discussion on why these source types were chosen. They are the same as in previous inventories, but is this a good choice? Can there improvements be made at this level already? Next it should be discussed what source types are most important, making big changes to a very small source type, is not going to change the total emissions significantly. Besides, emissions in China and Jiangsu in particular are dominated by solvent use and industrial processes, which is not the case in other regions and this needs to be pointed out early.

What is completely missing is a description of the data sources for the activity level data and a comparison to the previous emission inventories. It is not clear to me what was taken from the literature and what is really new here from this work in terms of activity data. These are essential for building this inventory and some of it is hinted on in Figure S3, but needs to be discussed here.

The discussion of the inventory development would benefit from re-organizing. I suggest discussing each source type in order and include activity factor, emission factor, speciation, source profile and uncertainty. As is, in chapters 2.2 to 2.5 the source types

are covered very inconsistent, which means that a lot of information is missing. All of this means some major re-organization of the first half of the paper, but without it cannot be judged what is new and important from this work. The rest of the paper was often difficult to follow, because it was not clear what all was included in the source type analysis.

Major Comments:

Page 6 line 125: Why did you choose these specific facilities? Were there no data on them?

Page 7 lines 146-150: Please explain why all these different stages were necessary for the GC analysis.

Page 9 line 207-212: The priorities for the emission factor determination need more detail. Are the emission limits for laws and regulations strictly enforced? If so, they can be used as an upper limit. If not, experience from other regions shows that these regulations rarely have much correlation with actual emissions. Also, what do you mean with expert judgment? Are those literature values or estimates from industry officials? Please explain!

Page 12 line 303: How much of the total emissions were covered with these measurements? Do they represent and help update much of the emissions? The source profile update in Figure 6 seems significant only for a few species?

Figure 6: The units in this Figure cannot be correct.

Page 14 line 353: Can you indicate how much the measurements changed the inventory? Were they an essential improvement compared to the previous inventories?

Page 15 line 382: I assume the units are g/kg?

Page 15 lines 396-405: Can you explain what "certain proxies" means?

Page 17 line 437-442: Please explain why the OFP to emissions ratio is important.

[Figure]

Page 17 lines 444-448: You need to provide the information about the most important species in the main text as well.

Page 17 lines 450-458: What are the most important species in the modeling shown later in the manuscript?

Page 17 lines 460-470: The text does not provide enough information on how the uncertainties are estimated, because of the many things that are not included in the inventory development section. It is therefore difficult to judge, if the emission factors are indeed the largest uncertainty. Can you indicate in this chapter, what would be the most helpful for improving the uncertainties. Would actual emissions measurements be critical?

Page 19: The omission of for example basic chemistry production in the activity data seems to be a clear indicator that the current emission inventory is more likely a lower limit of the emissions, because some of the activity is just not captured.

Page 19 line 520: What is your indication that the emissions in downtown are actually overestimated? It is clear that this method has a large uncertainty, but this does not necessarily mean an overestimation. Is this because the large point source emitters are not located in downtown and you include those in the downscaling method?

Page 21 line 556 (and elsewhere): Thylacetate is really not a very common chemical and I am surprised about the large atmospheric emissions. I think a discussion is warranted on what this compound is and why it is produced in such large quantities. I am wondering, if you meant ethyl acetate, which is pretty common?

Page 21 line 562-571: Overall the changes are not very large. What are the changes in OFP and total emissions and how much influence does the update have on ozone modeling?

Page 21 line 576: The qualification of the REAS inventory as extremely high should be removed. "Extremely high" is a very subjective term and should not be used. Please

be quantitative. Besides, it seems to me that this bottom up inventory provides more likely a lower limit for the emission, because there is a large potential for processes not being included. Also the modeling shows that ozone is still under-predicted, which also points to low emissions in this inventory.

Page 22 line 581: similar type of comment: "much larger emissions" is a very subjective term, so please be quantitative. I actually also disagree with the statement, the inventories are pretty close to each other, clearly within the stated uncertainties.

Chapter 4.4: I think the modeling shows that the inventory is still likely an underestimation of the actual emissions and I think this should be pointed out in the text.

Page 24 line 639: Please give a reference for the statement about VOC-limited.

Technical Comments:

Page 2 line 7: "field measurements of source profiles of the chemical industry. A total of 56 NMVOC samples"

Page 3 line 50: "during a heavy haze period"

Page 3 line54: "ozone formation was recognized"

Page 4 line 78: "6-18 times that of normal"

Page 5 line 95: "increasingly, domestic field measurements"

Page 5 line 104: "series of measurements have been conducted"

Page 6 line 133: "the canisters were made out of stainless steel"

Page 6 line 139: The sampling time was 10 minutes until the pressure in the canister reached ambient?

Page 6 line 140: " 50 meters downwind of the production"

Page 6 line 142: " a total of 56 samples"

Page 6 line 144: " NMVOC samples were analyzed"

Page 7 line 146: Firstly, the sample"

Page 13 line 337: SPECIATE

Page 13 line 344: " in which a solvent"

Page 15 line 392: emissions of motorcycles

Page 16 line 417: and other were between 26-30%

Page 21 line 560: cooking

Page 22 line 596: various

---

## Referee Comment (RC2) · Anonymous Referee #2 · 25 Mar 2017

Improved provincial emission inventory and speciation profiles of anthropogenic non-methane volatile organic compounds :a case study of Jiangsu, China By Zhao et al.

General comments:

The paper describes a bottom-up development and evaluation of a highly-resolved regional emission inventory for NMHCs in the area of Jiangsu, a strong industrialized region in Eastern China. The reference period is almost 10 years. The quantification of chemical processes is based on the determination of realistic source profiles for industrial activities by near-source measurements. The authors provide an extensive

work.

The paper is divided into 4 parts: first part describes the inventory methodology, second part compares the newly released emission inventory to other downscaled emission inventories regarding absolute emissions and speciated emissions; third part uses the CMAQ model to test the ability of the model to reproduce hourly maximum ozone concentrations with the new emission inventory. Evaluation of emission inventories is important for improving the simulation and forecast of air quality and climate and is unfortunately often neglected. While the science is of relevant atmospheric interest, I have major concerns about the paper:

1/ the paper is not easy to read and the reader gets easily lost. For instance, the authors often go backandforth with figures and associated discussion (ie. Figure 7). In several parts or sections, the paper relies on information reported in the Supplement Material which often makes the paper hard to follow, especially regarding the development of the emission inventory.

2/The sampling and analysis strategy in the field is not described and motivated. Line 103: the choice was put on the speciation of chemical industries. The sampling strategy and the representativeness of emission measurements should be detailed.

3/ Some sections do not provide reliable information

Lines 514-525: The authors compare the spatial distribution of emissions from industrial activities from three different methods including one without any information on individual plants, which uses proxies like population density. They show that the spatial allocation from this method is wrong and not representative of local characteristics. One could wonder whether such result could have been predictable.

4/ The interpretation of the figures are incomplete or approximate. Regarding the various emission inventory evaluation, it looks like the improvement with the update is not so clear in term of absolute concentration, reactivity and spatialization.
Lines 476-486: the discussion on uncertainty comparison should be revised or at least clarified. Differences in uncertainties between inventories in Table 3 could be also due to the way uncertainties are estimated or the spatial resolution. Indeed the authors give the impression that the uncertainty of the new released inventory is better. It might be for the wrong reason.

Line 551-561: Comparing the updated speciation of VOCs to the SPECIATE emission profiles is relevant. It is surprising to see that the updated VOC speciation profiles are not so different from the foreign SPECIATE database excepted aromatics and ethylacetate. It would be also relevant and interesting to compare database by only considering the measured profiles.

Figure 4: the comparison with other emission inventories for the same spatial domain reveals a quite good consistency regarding absolute values and trends. As stated by the authors all the results are within the 95% confidence limits. From this figure, it seems that differences are not statistically different. However the authors keep insisting on such differences. Figure 7: The difference between speciation is not so significant after updating and finally raises the question of the usefulness of the updating except for aldehydes. This deserves some discussion.

The simulation with the CMAQ model. The model outputs not only depend on the representation of emissions but also on the representation of chemistry and dynamics. The authors should take these two drivers as well to explain potential differences with the observed ozone-hourly maximum. Note that we also see some differences between seasons. Emissions might not be the only limitation. The get free from dilution effects or chemistry effects the authors could rather use ratios. To conclude. This paper is of importance and I would like to highlight the extensive work that has been accomplished. However, given my comments above, I would not recommend publication. I would encourage the author to submit again their manuscript after improving the clarity of the paper (organization) and the accuracy of conclusions. The authors should also reduce the length of the paper as some parts are not supportive.

Specific comments: Figure 2: characters are not visible Line 337: explain why SPECI-
ATE is used. There are other database (European) that might be also relevant. Please
explain why using SPECIATE. Line 439: explain the use and atmospheric relevancy of
the OFP/emission ratio
* * *

---

## Author Comment (AC1) · 7 May 2017

Manuscript No.: acp-2016-1121

Title: Improved provincial emission inventory and speciation profiles of anthropogenic non-methane volatile organic compounds: a case study for Jiangsu, China

Authors: Yu Zhao, Pan Mao, Yaduan Zhou, Yang Yang, Jie Zhang, Shekou Wang, Yanping Dong, Fangjian Xie, Yiyong Yu, Wenqing Li

We thank very much for the valuable comments and suggestions from reviewer 1,

which help us improve our manuscript significantly. The comments were carefully considered and revisions have been made in response to suggestions. Following is our point-by-point responses to the comments and corresponding revisions.

Reviewer #1

0. This paper describes the development and updates of a bottom-up regional anthropogenic emissions inventory for the Jiangsu province in China. Source profiles of industrial facilities were improved compared to previous inventories through measurements at the facility level of NMVOCs from canister samples. Differences to other inventories are discussed and regional CMAQ modeling studies with the various inventories were performed and it was found that using the current inventory improves, albeit still underestimates, ozone predictions. Improving anthropogenic emission inventories is the main pathway to improving air quality and climate modeling and forecasting, especially in areas such as China, which makes this paper a very valuable contribution. Unfortunately, the paper is difficult to read because of necessary information hidden in the SI, important data sources missing from the description and generally inconsistent description of the different source types and as a result this paper needs major changes, before it is acceptable.

Response and revisions:

We appreciate the reviewer's positive remarks on the contribution of the work. Regarding the weakness pointed out by the reviewer, we have improved the manuscript accordingly. Part of the information in the original supplement has been moved to the main text (please see details in our response to Q1), and data sources of emission inventory development have been clearly described in order as suggested by the reviewer (please see details in our response to Q3 and Q4). The description of different source types has been revised to avoid confusion. Please see the details in the following response and revision list to reviewer's comment.

1. My main issue with this paper is that the description of the inventory development

in chapters 2.2 and 2.3 is pretty unclear. First of all, the essential information of the source types is in the SI and not in the main text. A paper needs to be understandable even without reading the SI in detail and that is certainly not the case here. Tables S1 and S3 should be combined into one and moved to the main text. I also think that Tables S4, S5 and S6 should be combined and moved to the main text. Then the source types and the data sources for each type need to be discussed in more detail.

Response and revisions:

We thank and agree with the reviewer's comment. As suggested by the reviewer, we have moved Table S3 in the original supplement to the revised main manuscript (Table 1). The data sources of each type have been discussed in order in a new Section 2.3 in the revised manuscript. We have also merged Tables S4-S6 in the original supplement and moved the new table to the revised main manuscript (Table 2). Table S1 summarized the national emission estimates for the country by various studies, while the work focuses mainly on the provincial inventory. Therefore we kept Table S1 in the revised supplement.

2. I would like to see a discussion on why these source types were chosen. They are the same as in previous inventories, but is this a good choice? Can there improvements be made at this level already? Next it should be discussed what source types are most important, making big changes to a very small source type, is not going to change the total emissions significantly. Besides, emissions in China and Jiangsu in particular are dominated by solvent use and industrial processes, which is not the case in other regions and this needs to be pointed out early.

Response and revisions:

We thank the reviewer's comment. The framework of national and regional emission inventories was established based on China's energy and economic statistics. In this work, the framework for Jiangsu provincial inventory was updated and revised based on the plant-by-plant information from various data sources (including Environmental Statistics, Pollution Source Census, and on-site surveys as we stated in the text). Therefore the source categories were not "chosen" but were included in a framework that covered all the existing industrial processes for the provinces. We admit that there would be omission attributed to data limitation, but the omitted was supplemented as area sources. We have added the discussion in lines 177-190 in the revised manuscript.

For the importance of source types, we expected there were discrepancies in various provinces, and it could not be clearly explored unless detailed source information was incorporated. Therefore, we discussed the issue for Jiangsu in Section 3.2 in the revised manuscript where the emissions by source type were calculated and provided. At current stage, it is difficult to make a thorough emission comparison between studies, because there were not so many national and regional inventories which provided emissions and data sources at such a detailed source category level. We obtained available information and compared the results at coarser sector level in lines 674-693 in the revised manuscript. We also agree with the reviewer that emissions in China and Jiangsu in particular are dominated by solvent use and industrial processes, and we have stated the case early in lines 64-66 in the revised manuscript.

3. What is completely missing is a description of the data sources for the activity level data and a comparison to the previous emission inventories. It is not clear to me what was taken from the literature and what is really new here from this work in terms of activity data. These are essential for building this inventory and some of it is hinted on in Figure S3, but needs to be discussed here.

Response and revisions:

We thank the reviewer's comment. At the beginning of Section 2.2 (lines 177-188 in the revised manuscript), we first stressed the improvement in activity data compared to previous national and regional inventories: the inclusion of detailed plant-by-plant information through thorough investigations on available databases (i.e., the accuracy

of activity data for point sources were improved), more local information on traffic fleet, and the improved framework with more detailed classification of emission source categories. The subsequent changes in emission estimation and spatial distribution were then discussed in Section 4.1 and 4.3. We followed the reviewer's suggestion and provided detailed description on the data sources of activity levels by sector in the revised Section 2.3. However, we should admit that the methods of activity data collection from previous studies had also to be applied for certain sources (e.g., solvent use and oil distribution), as the whole framework of emission inventory was comprehensive and detailed activity data were currently unavailable for all the source categories at provincial (or smaller) scale.

4. The discussion of the inventory development would benefit from re-organizing. I suggest discussing each source type in order and include activity factor, emission factor, speciation, source profile and uncertainty. As is, in chapters 2.2 to 2.5 the source types are covered very inconsistent, which means that a lot of information is missing. All of this means some major re-organization of the first half of the paper, but without it cannot be judged what is new and important from this work. The rest of the paper was often difficult to follow, because it was not clear what all was included in the source type analysis.

Response and revisions:

We thank the reviewer's comment. Following the reviewer's suggestion, the section of inventory development has been re-organized. In the revised Section 2.2, the principle of emission inventory development was described including the methods of emission calculation, source profile estimation, and uncertainty analysis. In particular, at the beginning of Section 2.2 we have stressed the improvement in data sources compared to previous national and regional inventories. Original Sections 2.3-2.5 have been deleted in the revised manuscript. Instead, a new Section 2.3 has been added, which in order discussed the detailed methods and data sources of activity level estimation, data sources of emission factors and chemical profiles, and the probability distribution

functions (PDFs) by source category. We have also revised and updated the Tables 1 (Table S3 in the original supplement), 2 (Tables S4-6 in the original supplement), and S4 (Table S8 in the original supplement) to eliminate the inconsistency in source category description.

We need also to clarify a case in Figure 8, the comparison of sector emissions between different inventories. The sector classification for comparison between our work and Zhang et al. (2009)/MEIC was indeed inconsistent with other, because the latter two studies used a different classification. For direct comparison, therefore, we had to regroup the emissions of our work to be consistent with them in source categories. We have stated this in the revised caption of Figure 8.

5. Page 6 line 125: Why did you choose these specific facilities? Were there no data on them?

Response and revisions:

We thank the reviewer's important comment. Currently the chemical profiles of NMVOC emissions are still lacking for many source categories of chemical and refinery industry in China. The source types we selected for measurements were intensively distributed in Jiangsu province, and no domestic measurement has been conducted yet for those sources to our knowledge. Even in SPECIATE the chemical profiles were available for three processes, i.e., synthetic rubber, ethylene and polyethylene production. To improve the completeness of local source profiles of chemical industry, therefore, we chose those facilities for measurement. We have discussed this in lines 126-134 in the revised manuscript.

6. Page 7 lines 146-150: Please explain why all these different stages were necessary for the GC analysis.

Response and revisions:

We thank the reviewer's comment. In the first stage, vapour water was converted

to solid water and was thus separated from sample. In the second stage the target species were separated from $CO_2$ and other compositions in the air. The sample with target species was then concentrated in the third stage for GC analysis. We have added the explanation in lines 154-159 in the revised manuscript.

7. Page 9 line 207-212: The priorities for the emission factor determination need more detail. Are the emission limits for laws and regulations strictly enforced? If so, they can be used as an upper limit. If not, experience from other regions shows that these regulations rarely have much correlation with actual emissions. Also, what do you mean with expert judgment? Are those literature values or estimates from industry officials? Please explain!

Response and revisions:

We thank and agree the reviewer's comment. We (and previous studies) had to rely on emission limits of relevant regulations to estimate emissions for solvent use due mainly to lack of domestic test results. We have indicated in lines 323-326 in the revised manuscript that the regulations were not strictly enforced particularly for small solvent use enterprises and construction sites (area sources). Therefore, bias needs to be admitted here and possible underestimation in emissions would be expected for the sector. We have also clarified in lines 293-294 in the revised manuscript that emission factors from "expert judgment" indicated the data from routine investigations reported by the factory officials to local environmental protection bureaus.

8. Page 12 line 303: How much of the total emissions were covered with these measurements? Do they represent and help update much of the emissions? The source profile update in Figure 6 seems significant only for a few species?

Response and revisions:

We thank the reviewer's important comment. The source types we measured accounted for 9-11% of annual NMOVC emissions from chemical and refinery categories

(Oil exploitation and refinery, chemical raw materials, synthetic chemical industry, and fine chemical industry in Table 1) for Jiangsu 2005-2014. Attributed to a wide variety of source categories for chemical and refinery industry, the sources we measured did not dominate the emissions of the sector. However we mean the measurements improved the inventory (particularly for source profile) as there was no domestic measurement result on chemical composition published before on those sources, and it was the first time that local information was incorporated in the emission inventory development for those sources. In particular the contribution of the sources to emissions was enhanced in typical cities with intensive chemical industry. In Nanjing (the capital city of the province), for example, the source categories we measured accounted for 19% of annual emissions from chemical and refinery industry, and for 10% of the total anthropogenic emissions in 2011. Better information on chemical compositions of NMVOC emissions was expected to be available when the local measurements on source profile were conducted and incorporated for the city. Such information, however, would not be available until the entire emission inventory was developed. Therefore we added the information in lines 638-643 in the revised manuscript, after Section 3.2 that describes the NMVOC emissions by source type. Regarding Figure 6, as we described in lines 635-638 in the revised manuscript, both our measurements and other domestic studies after 2010 were incorporated for source profile updating. We have stated that relatively big changes in emissions (over 10 Gg) were found for certain compositions including ethylacetate and aromatics species (benzene, xylene, ethyl benzene, and methyl benzene). To clarify the impact, we further calculated the total difference between emissions of all species before and after updating, and it reached 281 Gg, i.e., 13% of the total emissions for the province in 2010. We have added the discussion in lines 646-648 in the revised manuscript.

9. Figure 6: The units in this Figure cannot be correct

Response and revisions:

We thank the reviewer's reminder and the error has been corrected in the revised

[Figure]

Figure 6.

10. Page 14 line 353: Can you indicate how much the measurements changed the inventory? Were they an essential improvement compared to the previous inventories?

Response and revisions:

We thank the reviewer's comment. As the measurements targeted on the source profile (i.e., the fractions of chemical species), incorporation of the results could hardly lead to changes in total NMVOC emission estimation. However, as we responded to Q8, the source types we measured accounted for 9-11% of NMVOC emissions from chemical and refinery categories in the province, and the fraction could be even larger in typical city. Better information on NMVOC emission speciation could thus be available when the measurement results were applied for source profile updating compared to previous inventories. The improvement could also be evaluated through air quality modeling for certain species (e.g., ozone) as we presented in Section 4.4, although further analysis is still needed (please also see our response to Q15).

We also need to acknowledge the big variety of source categories for chemical and refinery industry in China. At current stage measurements on chemical sources are still lacking for many source types. Although incremental contribution could be expected through current work, more efforts on field measurments from different research groups are still needed in order to obtain a more complete database of chemical profiles for China's chemical and refinery industry. We have added the discussion in lines 655-661 in the revised manuscript.

11. Page 15 line 382: I assume the units are g/kg?

Response and revisions:

We thank the reviewer's reminder and the error has been corrected in the revised manuscript.

12. Page 15 lines 396-405: Can you explain what "certain proxies" means?

Response and revisions:

"Certain proxies" indicate the parameters used for allocating the emissions from sources other than point sources, including GDP, population, road net and traffic flow, and railway and canal net. We revised the sentence in lines 648-671 in the revised manuscript to make it clearer: "For other sources, certain proxies were applied to allocate emissions, including GDP for industrial area sources and oil distribution, population for solve use area sources, road net and traffic flow for on-road vehicles, railway and canal net for off-road transportation, and rural population for biomass burning."

13. Page 17 line 437-442: Please explain why the OFP to emissions ratio is important.

Response and revisions:

OFP is used for evaluating the capability of ozone formation through atmospheric chemical reactions for individual VOC species. As the chemical profiles of emitted VOC vary between source categories, the OFP to emission ratio for a given sector (or source type) indicates the potential contribution to ozone formation for the sector (or source type), as a combined effects of multiple NMVOC species emitted from it. The ratio could thus provide scientific suggestion of emission control for policy makers, e.g., the emission control needs to be preferentially considered for sectors with large OFP/emissions. We have explained the issue in lines 513-517 in the revised manuscript.

14. Page 17 lines 444-448: You need to provide the information about the most important species in the main text as well.

Response and revisions:

We thank the reviewer's reminder. In lines 526-527 in the revised manuscript we have added the information as required: "Xylene, ethylene, and propylene were identified as the most three important species in terms of OFP."

15. Page 17 lines 450-458: What are the most important species in the modeling

shown later in the manuscript?

Response and revisions:

We thank the reviewer's important comment. In lines 750-752 in the revised manuscript, we have indicated that the updated provincial inventory provided larger emission estimates for certain species with relatively high ozone formation potential including ethene and ethanol. Such revisions were expected to improve the ozone simulation with CMAQ as we described in Section 4.4. We should admit, however, that the impacts of emission changes for individual species on ozone simulation could not be completely confirmed in current work. The improved ozone simulation presented here was a combined effects of an updated inventory with revisions on emission estimation, spatial distribution and source profiles for all relevant species. A much more detailed chemistry transport modeling study with intensive sensitivity analysis is needed in order to further figure out the impacts of individual species (as well as factors other than emission input). We mean it is beyond the scope of the current work, and we will keep conducting the relevant analysis in the future. We have discussed this in lines 752-756 in the revised manuscript.

16. Page 17 lines 460-470: The text does not provide enough information on how the uncertainties are estimated, because of the many things that are not included in the inventory development section. It is therefore difficult to judge, if the emission factors are indeed the largest uncertainty. Can you indicate in this chapter, what would be the most helpful for improving the uncertainties. Would actual emissions measurements be critical?

Response and revisions:

We thank and agree the reviewer's comment. As we respond to Q4 of the reviewer, we have reorganized the inventory development section carefully, and more detailed description on uncertainty analysis has been added into the revised manuscript. We have briefly described the principles of uncertainty quantification in lines 235-242, Section

2.2, and we have added the detailed methods and data sources of determination of probability distribution function (PDF) for emission factors and activity levels by source category in Section 2.3. In lines 273-277 in the revised manuscript, we have admitted that uncertainty of emission factor was evaluated depending on expert judgment due to insufficient data support from local measurments, and that PDF of emission factor was given according to reliability of data sources and/or the robustness of calculation methods following the rules of previous work (Streets et al., 2003; Wei et al., 2011). As suggested by the reviewer, we have indicated in lines 552-554, Section 3.4 of the revised manuscript that actual emissions measurements would be helpful for reducing the uncertainty, since expanded data samples would better support the determination of emission factors through data fitting instead of conservative assumption.

17. Page 19: The omission of for example basic chemistry production in the activity data seems to be a clear indicator that the current emission inventory is more likely a lower limit of the emissions, because some of the activity is just not captured.

Response and revisions:

We thank the reviewer's comment. As we stated, the basic chemistry production was not captured in economic statistics, upon which the Method 3 was based. Method 3 was actually applied by most previous national and regional inventories attributed mainly to lack of detailed emission source data, and underestimation in emissions can be expected as judged by the reviewer. In this work, Method 1 that incorporated the most available information from Environmental Statistics, Pollution Source Census, and on-site surveys was applied, thus the errors from activity data omission for certain sources were corrected. We mean the comparisons between different methods just highlighted the improved estimation in emissions by this work. In lines 578-581 in the revised manuscript, we stressed that Method 1 was applied in this work to provide the best available emission information. In lines 598-600 in the revised manuscript, we also admitted that even Method 1 could lead to emission underestimation as it is difficult to cover all the process in emission factor determination, particularly for the

fugitive release.

18. Page 19 line 520: What is your indication that the emissions in downtown are actually overestimated? It is clear that this method has a large uncertainty, but this does not necessarily mean an overestimation. Is this because the large point source emitters are not located in downtown and you include those in the downscaling method?

Response and revisions:

We thank and agree the reviewer's comment. We mean the uncertainty came mainly from the method of spatial allocation of emissions. Attributed to lack of detailed information of emission source locations, Method 3 in general allocated the emissions based on the density of economy and population, and it would thus lead to larger emission estimation in downtown area with relatively high density of economic activities and population, compared to suburban area. In this case of Nanjing, however, the large point sources (e.g., chemical industry and refinery plants) were not located in downtown, and discrepancy between the estimation and actual emission distribution would occur. As the total emissions in Method 3 could be underestimated due to omission of certain source categories, we agree with the reviewer that the absolute emission levels might not be necessarily overestimated, and we revised the words as "would overestimate the fraction of emissions in urban downtown" in line 607 in the revised manuscript.

19. Page 21 line 556 (and elsewhere): Thylacetate is really not a very common chemical and I am surprised about the large atmospheric emissions. I think a discussion is warranted on what this compound is and why it is produced in such large quantities. I am wondering, if you meant ethyl acetate, which is pretty common?

Response and revisions:

We thank the reviewer's reminder and admitted the typo error. Yes it should be ethyl acetate, and we have corrected it in the revised manuscript.

20. Page 21 line 562-571: Overall the changes are not very large. What are the changes in OFP and total emissions and how much influence does the update have on ozone modeling?

Response and revisions:

We thank the reviewer's reminder. The relative changes actually varied among species and could be big for certain species. Please see the detailed data in the attached table. The total NMVOC emissions estimated in this work was 316 Gg (i.e., 18%) larger than MEIC for 2010. We have added the information in lines 695-697 in the revised manuscript. Unfortunately the OFP for MEIC could not be calculated as the original information of chemical species was unavailable. The influence of updated emission inventory on ozone modeling was described in Section 4.4. As we responded to Q15, current work could not totally disentangle the effects of source profile updating and other changes in emission inventory (e.g., total amount and spatial distribution of emissions), and we will continue the research in the future.

21. Page 21 line 576: The qualification of the REAS inventory as extremely high should be removed. "Extremely high" is a very subjective term and should not be used. Please be quantitative. Besides, it seems to me that this bottom up inventory provides more likely a lower limit for the emission, because there is a large potential for processes not being included. Also the modeling shows that ozone is still under-predicted, which also points to low emissions in this inventory.

Response and revisions: We thank and agree the reviewer's comment. We have revised the text with quantitative comparison "Except for REAS that provided 37%-77% higher emissions than this work for 2005-2008" in line 677 in the revised manuscript. As we response to Q17, the errors from activity data omission for certain sources were corrected in the updated provincial inventory in this work. Nevertheless, we agree with the reviewer that the under-predicted ozone still implied the possible underestimation in VOC emissions. We have stressed this in lines 743-746 in the revised manuscript.

[Figure]

22. Page 22 line 581: similar type of comment: "much larger emissions" is a very subjective term, so please be quantitative. I actually also disagree with the statement, the inventories are pretty close to each other, clearly within the stated uncertainties.

Response and revisions:

We thank and agree the reviewer's comment. Subject term has been deleted and quantitative information has been provided as "the emissions in this work were 4% and 20% larger than the national inventory for 2005 (Wei et al., 2008) and regional inventory for 2010 (Fu et al., 2013), respectively" in lines 682-684 in the revised manuscript.

23. Chapter 4.4: I think the modeling shows that the inventory is still likely an underestimation of the actual emissions and I think this should be pointed out in the text.

Response and revisions:

We thank and agree the reviewer's comment. In lines 743-746 in the revised manuscript, we have stated "the updated anthropogenic NMVOC emission inventory at provincial scale was still likely an underestimation of the actual emissions".

24. Page 24 line 639: Please give a reference for the statement about VOC-limited.

Response and revisions:

We thank the reviewer's reminder and a reference (Xing et al., 2011) has been given in the revised manuscript.

25. Technical Comments

Response and revisions:

We thank the reviewer's reminder and all the corrections have been made as suggested by the reviewer.

References

Fu, X., Wang, S., Zhao, B., Xing, J., Cheng, Z., Liu, H., and Hao, J.: Emission inventory of primary pollutants and chemical speciation in 2010 for the Yangtze River Delta region, China, Atmos. Environ., 70, 39-50, 2013.

Streets, D. G., Bond, T. C., Carmichael, G. R., Fernandes, S. D., Fu, Q., He, D., Klimont, Z., Nelson, S. M., Tsai, N. Y., Wang, M. Q., Woo, J. H., and Yarber, K. F.: An inventory of gaseous and primary aerosol emissions in Asia in the year 2000, J. Geophys. Res., 108 (D21), 8809, doi: 10.1029/2002jd003093, 2003.

Wei, W., Wang, S., Chatani, S., Klimont, Z., Cofala, J. and Hao, J.: Emission and speciation of non-methane volatile organic compounds from anthropogenic sources in China, Atmos. Environ., 42, 4976-4988, 2008.

Wei, W., Wang, S., and Hao, J.: Uncertainty analysis of emission inventory for volatile organic compounds from anthropogenic source in China, Environmental Science, 32, 305-312 (in Chinese). Xing, J., Wang, S. X., Jang, C., Zhu, Y. and Hao, J. M.: Nonlinear response of ozone to precursor emission changes in China: a modeling study using response surface methodology, Atmos. Chem. Phys., 11, 5027-5044, 2011.

Zhang, Q., Streets, D. G., Carmichael, G. R., He, K. B., Huo, H., Kannari, A., Klimont, Z., Park, I. S., Reddy, S., Fu, J. S., Chen, D., Duan, L., Lei, Y., Wang, L. T., and Yao, Z. L.: Asian emissions in 2006 for the NASA INTEX-B mission, Atmos. Chem. Phys., 9, 5131-5153, 2009.

[Figure]

Table R1 Emissions of CB05 species for Jiangsu 2010

| Inventory | Source | Emission ($10^9$ mol) | | | | | | | | | | | | | |
|---|---|---|---|---|---|---|---|---|---|---|---|---|---|---|---|
| | | PAR | OLE | TOL | XYL | FORM | ALD2 | ETH | MEOH | ETOH | ETHA | IOLE | ALDX | UNR | NVOL |
| Before updating | Fossil fuel combustion | 1.00 | 0.08 | 0.06 | 0.10 | 0.00 | 0.00 | 0.17 | 0.00 | 0.00 | 0.15 | 0.02 | 0.00 | 0.35 | 0.00 |
| | Industrial process | 17.19 | 0.93 | 0.73 | 0.06 | 0.01 | 0.03 | 1.52 | 0.20 | 0.48 | 0.96 | 0.08 | 0.01 | 5.54 | 0.00 |
| | Transportation | 10.84 | 0.53 | 0.21 | 0.25 | 0.28 | 0.26 | 0.60 | 0.00 | 0.00 | 0.13 | 0.21 | 0.18 | 0.78 | 0.00 |
| | Solvent use | 12.89 | 0.18 | 1.17 | 0.35 | 0.06 | 0.01 | 0.03 | 0.02 | 0.03 | 0.02 | 0.06 | 0.00 | 2.18 | 0.00 |
| | Oil distribution | 2.75 | 0.02 | 0.01 | 0.00 | 0.02 | 0.02 | 0.00 | 0.00 | 0.00 | 0.02 | 0.04 | 0.00 | 0.11 | 0.00 |
| | Biomass burning | 1.69 | 0.24 | 0.02 | 0.12 | 0.31 | 0.14 | 0.40 | 0.31 | 0.00 | 0.18 | 0.02 | 0.09 | 0.81 | 0.00 |
| | Other | 1.34 | 0.13 | 0.00 | 0.00 | 0.29 | 0.34 | 0.13 | 0.00 | 0.00 | 0.10 | 0.00 | 0.42 | 0.24 | 0.00 |
| | Total | 47.70 | 2.10 | 2.20 | 0.88 | 0.97 | 0.81 | 2.84 | 0.54 | 0.52 | 1.56 | 0.44 | 0.69 | 10.01 | 0.01 |
| After updating | Fossil fuel combustion | 1.00 | 0.08 | 0.06 | 0.10 | 0.00 | 0.00 | 0.17 | 0.00 | 0.00 | 0.15 | 0.02 | 0.00 | 0.35 | 0.00 |
| | Industrial process | 18.42 | 0.92 | 0.55 | 0.14 | 0.04 | 0.03 | 1.28 | 0.18 | 0.49 | 1.05 | 0.12 | 0.12 | 4.59 | 0.00 |
| | Transportation | 11.03 | 0.48 | 0.21 | 0.27 | 0.38 | 0.17 | 0.60 | 0.00 | 0.00 | 0.12 | 0.13 | 0.26 | 0.91 | 0.00 |
| | Solvent use | 14.95 | 0.20 | 1.01 | 0.39 | 0.05 | 0.01 | 0.21 | 0.01 | 0.04 | 0.15 | 0.07 | 0.59 | 2.07 | 0.00 |
| | Oil distribution | 2.75 | 0.02 | 0.01 | 0.00 | 0.02 | 0.02 | 0.00 | 0.00 | 0.00 | 0.02 | 0.04 | 0.00 | 0.11 | 0.00 |
| | Biomass burning | 1.79 | 0.23 | 0.03 | 0.12 | 0.23 | 0.14 | 0.37 | 0.23 | 0.00 | 0.18 | 0.02 | 0.18 | 0.77 | 0.00 |
| | Other | 1.34 | 0.13 | 0.00 | 0.00 | 0.29 | 0.34 | 0.13 | 0.00 | 0.00 | 0.10 | 0.00 | 0.42 | 0.24 | 0.00 |
| | Total | 51.29 | 2.06 | 1.87 | 1.05 | 1.00 | 0.72 | 2.75 | 0.42 | 0.54 | 1.77 | 0.40 | 1.57 | 9.04 | 0.01 |
| MEIC | - | 47.31 | 2.08 | 3.49 | 1.85 | 1.10 | 0.49 | 1.99 | 0.48 | 0.30 | 1.12 | 0.55 | 1.24 | 9.25 | 0.74 |
| Difference between after updating and MEIC(relative to MEIC) | | 8.4% | -1.2% | -46.4% | -43.3% | -8.6% | 47.7% | 38.4% | -12.1% | 77.1% | 58.7% | -27.5% | 27.4% | -2.3% | -99.3% |

**Fig. 1.**

---

## Author Comment (AC2) · 7 May 2017

Manuscript No.: acp-2016-1121

Title: Improved provincial emission inventory and speciation profiles of anthropogenic non-methane volatile organic compounds: a case study for Jiangsu, China Authors: Yu Zhao, Pan Mao, Yaduan Zhou, Yang Yang, Jie Zhang, Shekou Wang, Yanping Dong, Fangjian Xie, Yiyong Yu, Wenqing Li

We thank very much for the valuable comments and suggestions from reviewer 2, which help us improve our manuscript significantly. The comments were carefully con-

sidered and revisions have been made in response to suggestions. Following is our point-by-point responses to the comments and corresponding revisions.

Reviewer #2

0. The paper describes a bottom-up development and evaluation of a highly-resolved regional emission inventory for NMHCs in the area of Jiangsu, a strong industrialized region in Eastern China. The reference period is almost 10 years. The quantification of chemical processes is based on the determination of realistic source profiles for industrial activities by near-source measurements. The authors provide an extensive work. The paper is divided into 4 parts: first part describes the inventory methodology, second part compares the newly released emission inventory to other downscaled emission inventories regarding absolute emissions and speciated emissions; third part uses the CMAQ model to test the ability of the model to reproduce hourly maximum ozone concentrations with the new emission inventory. Evaluation of emission inventories is important for improving the simulation and forecast of air quality and climate and is unfortunately often neglected. Response and revisions:

We appreciate the reviewer's remarks on the importance of the work.

1. While the science is of relevant atmospheric interest, I have major concerns about the paper: the paper is not easy to read and the reader gets easily lost. For instance, the authors often go back and forth with figures and associated discussion (i.e., Figure 7). In several parts or sections, the paper relies on information reported in the Supplement Material which often makes the paper hard to follow, especially regarding the development of the emission inventory.

Response and revisions:

We thank the reviewer's comment. In the revised manuscript, we moved important information in the original supplement to the main text, including Table S3 that illustrated the framework of emission inventory development and classification of emission source

categories (Table 1 in the revised manuscript), and Table S4-S6 that summarized the emission factors by source category (a merged Table 2 in the revised manuscript). The section of emission inventory development has been re-organized. The revised Section 2.2 described the principles of emission inventory development including the methods of emission calculation, source profile estimation, and uncertainty analysis. In particular, the general improvement in data sources compared to previous national and regional inventories was stressed at the beginning of the section. Original Sections 2.3-2.5 have been deleted in the revised manuscript. Instead, a new Section 2.3 has been added, which in order discussed the detailed methods and data sources of activity level estimation, data sources of emission factors and chemical profiles, and the probability distribution functions (PDFs) by source category. We have also moved the original Table 1 to Section 3.3 (Table 4 in the revised manuscript) as it actually described the results of chemical profiles by source category. We expect such revisions make the text clearer and easier to read. To be concise, Figure 7 illustrated the distributions of chemical species under different mechanisms both in this work and in other inventories, thus we had to discuss the figure in the speciation section (Section 4.2) and comparison section (Section 4.3). We have revised the text in lines 665-666 in the revised manuscript, and hopefully it would help to clarify the case.

2. The sampling and analysis strategy in the field is not described and motivated. Line 103: the choice was put on the speciation of chemical industries. The sampling strategy and the representativeness of emission measurements should be detailed.

Response and revisions:

We thank the reviewer's important comment. Currently the chemical profiles of NMVOC emissions are still lacking for many source categories of chemical and refinery industry in China. The source types we selected for measurements were intensively distributed in Jiangsu province, and no domestic measurement has been conducted yet for those sources to our knowledge. Even in SPECIATE the chemical profiles were available for three processes, i.e., synthetic rubber, ethylene and polyethylene production. To

improve the completeness of local source profiles of chemical industry, therefore, we chose those source categories for measurement. We have discussed this in lines 126-134 in the revised manuscript.

3. Some sections do not provide reliable information. Lines 514-525: The authors compare the spatial distribution of emissions from industrial activities from three different methods including one without any information on individual plants, which uses proxies like population density. They show that the spatial allocation from this method is wrong and not representative of local characteristics. One could wonder whether such result could have been predictable.

Response and revisions:

We thank the reviewer's comment. The motivation of the comparison was to reveal the discrepancies in emission estimation and spatial distribution between inventories with different data sources. In actual, the method with little information on individual plants, as mentioned by the reviewer, was commonly applied in national and regional inventories at larger spatial scales, and emission downscaling was generally adopted when high-resolution inventory was needed for air quality simulation or other purposes. We have pointed this in line 583 in the revised manuscript. Through the comparison conducted in this work, we illustrated that such method could lead to big uncertainty in allocation of emissions and thereby air quality simulation, at least for cities like Nanjing whose emissions were dominated by big industrial plants. Therefore we mean the work highlighted the necessity of careful investigation on individual emission sources when the accuracy of local inventory became a big concern for both scientific community and China's policy makers of air pollution control.

4. The interpretation of the figures are incomplete or approximate. Regarding the various emission inventory evaluation, it looks like the improvement with the update is not so clear in term of absolute concentration, reactivity and spatialization.

Response and revisions:

We thank the reviewer's comment. In the revised manuscript, we have provided more quantitative information in the interpretation of figures, particularly for Figures 4, 6, 7, and 8. The differences in emission estimation and chemical profiles between various inventories have been stressed with detailed data support. Subject terms have also been deleted to avoid confusion. Please check the revisions in lines 646-661, 676-684, and 695-700 in the revised manuscript.

5. Lines 476-486: the discussion on uncertainty comparison should be revised or at least clarified. Differences in uncertainties between inventories in Table 3 could be also due to the way uncertainties are estimated or the spatial resolution. Indeed the authors give the impression that the uncertainty of the new released inventory is better. It might be for the wrong reason.

Response and revisions:

We thank and agree the reviewer's comment. Among all the studies we include for comparison, Huang et al. (2011) applied a different method to calculate the uncertainty other than Monte-Carlo simulation, and it would lead to difference in uncertainty estimation. We have added the discussion in lines 559-564 in the revised manuscript. Regarding the spatial scale, as current study focused only on the total emissions, the discrepancies in resolution (and thereby the uncertainty in spatial distribution of emissions) were not covered here. We agree with the reviewer that uncertainties could differ for inventories at different spatial scales. We expect such difference resulted mainly from the various levels of details for emission source information, as we have discussed in the section.

6. Line 551-561: Comparing the updated speciation of VOCs to the SPECIATE emission profiles is relevant. It is surprising to see that the updated VOC speciation profiles are not so different from the foreign SPECIATE database excepted aromatics and ethylacetate. It would be also relevant and interesting to compare database by only considering the measured profiles.

Response and revisions:

We thank the reviewer's comment. We admit there was a confusion in the original text. The source profile before updating was not directly taken from SPECIATE but a combination of Li et al. (2014) and SPECIATE. Li et al. (2014) made a comprehensive source profile for China based on domestic measurement results published before 2010. As a more detailed source classification was used in this work, some sources were not covered by Li et al. (2014) and thus the results from SPECIATE were applied. In this work, as we indicated in Sections 2.2 and 2.3 of the revised manuscript, the domestic measurements after 2010 and the local measurements by us were incorporated to update the source profile. As the newly-added measurements were not sufficient enough, not very big differences were found between source profiles between and after updating except for aromatics and ethylacetate.

We have clarified this in lines 211-224 in the revised manuscript. Among the sources we measured, the chemical profiles of synthetic rubber, ethylene and polyethylene production were available in SPECIATE. As suggested by the reviewer, therefore, we have compared the profiles between our results and SPECIATE for the three source categories, as shown in Figure 1b-d. Relevant discussions have been provided in lines 405-422 in the revised manuscript.

7. Figure 4: the comparison with other emission inventories for the same spatial domain reveals a quite good consistency regarding absolute values and trends. As stated by the authors all the results are within the 95% confidence limits. From this figure, it seems that differences are not statistically different. However the authors keep insisting on such differences.

Response and revisions:

We thank and agree with the reviewer's comment. We have deleted the subject term and conducted quantitative comparisons in total emissions between different inventories in lines 676-684 in the revised manuscript. We admit that the differences between

our study and most other inventories were not large except for REAS. As shown in Figure 8, however, emissions of different sectors varied between studies attributed to the various data sources and calculation methods. Therefore, we kept the analysis on the different emissions by sector, and tried to reveal the effects of data sources on NMVOC emission estimation for typical source categories.

8. Figure 7: The difference between speciation is not so significant after updating and finally raises the question of the usefulness of the updating except for aldehydes. This deserves some discussion.

Response and revisions:

We thank the reviewer's comment. The relative changes actually varied among species and could be big for certain species. Please see the detailed data in the attached table. For example, the updated provincial inventory provided larger emission estimates for ethene and ethanol with relatively high ozone formation. We have added the information in lines 695-700 in the revised manuscript. As we indicated in the section 4.4 (CMAQ evaluation), the updated speciation of NMVOC emissions were expected to improve the ozone simulation with chemistry transport modeling. However, we also admitted that the improved ozone simulation was a combined effects of an updated inventory with revisions on total emission estimation, spatial distribution and source profiles for all relevant species. Current work could not totally disentangle the effects of source profile updating and other changes in emission inventory. A detailed chemistry transport modeling study with sensitivity analysis is needed in order to further figure out the impacts of individual species, and we will keep conducting the relevant analysis in the future. We have discussed this in lines 752-756 in the revised manuscript.

9. The simulation with the CMAQ model. The model outputs not only depend on the representation of emissions but also on the representation of chemistry and dynamics. The authors should take these two drivers as well to explain potential differences with the observed ozone-hourly maximum. Note that we also see some differences between

seasons. Emissions might not be the only limitation. The get free from dilution effects or chemistry effects the authors could rather use ratios.

Response and revisions:

We thank and agree the reviewer's important comment. Since the current work focused mainly on the emission inventory revision and its subsequent impacts on air quality modeling, the model configurations were same on parameters other than emission input, including the chemistry mechanisms, meteorology condition and dynamics as mentioned by the reviewer. Therefore we believe that the discrepancy in ozone concentrations between the two simulations using MEIC and updated provincial inventory came most from the varied estimations on emission level, source profiles, and spatial distribution of emissions. We have stated this in lines 733-735 in the revised manuscript. We also agree with the reviewer that emission input was not the only limitation in chemistry transport modeling. To further figure out the impacts of chemistry and dynamics on ozone simulation, however, a more detailed sensitivity analysis will be needed on relevant parameters. We mean it is beyond the scope of the current work, and we will keep conducting the relevant analysis in the future. We have discussed such limitation in lines 765-767 in the revised manuscript.

10. Specific comments: Figure 2: characters are not visible; Line 337: explain why SPECIATE is used. There are other database (European) that might be also relevant. Please explain why using SPECIATE. Line 439: explain the use and atmospheric relevancy of the OFP/emission ratio

Response and revisions: We thank reviewer's comment and reminder.

The characters in Figure 2 are now visible.

As some manufacturing technologies we measured are quite unique in China, few test results for the same source types were reported in European databases such as Theloke and Friedirch (2007), thus direct comparison was not conducted.

[Figure]

OFP is used for evaluating the capability of ozone formation through atmospheric chemical reactions for individual VOC species. As the chemical profiles of emitted VOC vary between source categories, the OFP to emission ratio for a given sector (or source type) indicates the potential contribution to ozone formation for the sector (or source type), as a combined effects of multiple NMVOC species emitted from it. The ratio could thus provide scientific suggestion of emission control for policy makers, e.g., the emission control needs to be preferentially considered for sectors with large OFP/emissions. We have explained this issue in lines 513-517 in the revised manuscript.

11. To conclude. This paper is of importance and I would like to highlight the extensive work that has been accomplished. However, given my comments above, I would not recommend publication. I would encourage the author to submit again their manuscript after improving the clarity of the paper (organization) and the accuracy of conclusions. The authors should also reduce the length of the paper as some parts are not supportive.

Response and revisions:

Again we appreciate the reviewer's remarks on the importance of the work. Regarding the weakness pointed out by the reviewer, we have improved the manuscript accordingly. In particular we have reorganized the text and provided more detailed information on emission inventory development to make the manuscript clear. Important data have been moved from the original supplement to the main text. The discussion of results has also been improved to avoid subjective statement and confusion, according to the reviewer's comments and suggestion. Please see the details in the response and revision list to the reviewer's comment. We have also tried our best to be concise in text. As much detailed information on methods and data sources (including relevant tables) has been added into the revised manuscript, however, the length of paper could hardly be reduced. We hope the revision could meet the quality standard of publication in Atmos. Chem. Phys.

**References**

Huang, C., Chen, C. H., Li, L., and Cheng, Z.: Emission inventory of anthropogenic air pollutants and VOC species in the Yangtze River Delta region, China, Atmos. Chem. Phys., 11, 4105-4120, 2011.

Li, M., Zhang, Q., Streets, D. G., He, K. B., Cheng, Y. F., Emmons, L. K., Huo, H., Kang, S. C., Lu, Z., Shao, M., Su, H., Yu, X., and Zhang, Y.: Mapping Asian anthropogenic emissions of non-methane volatile organic compounds to multiple chemical mechanisms, Atmos. Chem. Phys., 14, 5617-5638, 2014.

Theloke, J., and Friedrich, R.: Compilation of a database on the composition of anthropogenic VOC emissions for atmospheric modeling in Europe, Atmos. Environ., 41, 4148-4160, 2007.

[Figure]

Table R2 Emissions of CB05 species for Jiangsu 2010

| Inventory | Source | Emission (10⁹ mol) | | | | | | | | | | | | | |
|---|---|---|---|---|---|---|---|---|---|---|---|---|---|---|---|
| | | PAR | OLE | TOL | XYL | FORM | ALD2 | ETH | MEOH | ETOH | ETHA | IOLE | ALDX | UNR | NVOL |
| Before updating | Fossil fuel combustion | 1.00 | 0.08 | 0.06 | 0.10 | 0.00 | 0.00 | 0.17 | 0.00 | 0.00 | 0.15 | 0.02 | 0.00 | 0.35 | 0.00 |
| | Industrial process | 17.19 | 0.93 | 0.73 | 0.06 | 0.01 | 0.03 | 1.52 | 0.20 | 0.48 | 0.96 | 0.08 | 0.01 | 5.54 | 0.00 |
| | Transportation | 10.84 | 0.53 | 0.21 | 0.25 | 0.28 | 0.26 | 0.60 | 0.00 | 0.00 | 0.13 | 0.21 | 0.18 | 0.78 | 0.00 |
| | Solvent use | 12.89 | 0.18 | 1.17 | 0.35 | 0.06 | 0.01 | 0.03 | 0.02 | 0.03 | 0.02 | 0.06 | 0.00 | 2.18 | 0.00 |
| | Oil distribution | 2.75 | 0.02 | 0.01 | 0.00 | 0.02 | 0.02 | 0.00 | 0.00 | 0.00 | 0.02 | 0.04 | 0.00 | 0.11 | 0.00 |
| | Biomass burning | 1.69 | 0.24 | 0.02 | 0.12 | 0.31 | 0.14 | 0.40 | 0.31 | 0.00 | 0.18 | 0.02 | 0.09 | 0.81 | 0.00 |
| | Other | 1.34 | 0.13 | 0.00 | 0.00 | 0.29 | 0.34 | 0.13 | 0.00 | 0.00 | 0.10 | 0.00 | 0.42 | 0.24 | 0.00 |
| | Total | 47.70 | 2.10 | 2.20 | 0.88 | 0.97 | 0.81 | 2.84 | 0.54 | 0.52 | 1.56 | 0.44 | 0.69 | 10.01 | 0.01 |
| After updating | Fossil fuel combustion | 1.00 | 0.08 | 0.06 | 0.10 | 0.00 | 0.00 | 0.17 | 0.00 | 0.00 | 0.15 | 0.02 | 0.00 | 0.35 | 0.00 |
| | Industrial process | 18.42 | 0.92 | 0.55 | 0.14 | 0.04 | 0.03 | 1.28 | 0.18 | 0.49 | 1.05 | 0.12 | 0.12 | 4.59 | 0.00 |
| | Transportation | 11.03 | 0.48 | 0.21 | 0.27 | 0.38 | 0.17 | 0.60 | 0.00 | 0.00 | 0.12 | 0.13 | 0.26 | 0.91 | 0.00 |
| | Solvent use | 14.95 | 0.20 | 1.01 | 0.39 | 0.05 | 0.01 | 0.21 | 0.01 | 0.04 | 0.15 | 0.07 | 0.59 | 2.07 | 0.00 |
| | Oil distribution | 2.75 | 0.02 | 0.01 | 0.00 | 0.02 | 0.02 | 0.00 | 0.00 | 0.00 | 0.02 | 0.04 | 0.00 | 0.11 | 0.00 |
| | Biomass burning | 1.79 | 0.23 | 0.03 | 0.12 | 0.23 | 0.14 | 0.37 | 0.23 | 0.00 | 0.18 | 0.02 | 0.18 | 0.77 | 0.00 |
| | Other | 1.34 | 0.13 | 0.00 | 0.00 | 0.29 | 0.34 | 0.13 | 0.00 | 0.00 | 0.10 | 0.00 | 0.42 | 0.24 | 0.00 |
| | Total | 51.29 | 2.06 | 1.87 | 1.05 | 1.00 | 0.72 | 2.75 | 0.42 | 0.54 | 1.77 | 0.40 | 1.57 | 9.04 | 0.01 |
| MEIC | - | 47.31 | 2.08 | 3.49 | 1.85 | 1.10 | 0.49 | 1.99 | 0.48 | 0.30 | 1.12 | 0.55 | 1.24 | 9.25 | 0.74 |
| Difference between after updating and MEIC(relative to MEIC) | | 8.4% | -1.2% | -46.4% | -43.3% | -8.6% | 47.7% | 38.4% | -12.1% | 77.1% | 58.7% | -27.5% | 27.4% | -2.3% | -99.3% |

**Fig. 1.**